**Investigation**

# Differentiating mechanism from outcome for ancestry-assortative mating in admixed human populations

Dashiell J. Massey [iD],[1] Zachary A. Szpiech [iD],[2] Amy Goldberg [iD][1,*]

[1]Department of Evolutionary Anthropology, Duke University, Durham, NC 27708, USA
[2]Department of Biology, The Pennsylvania State University, University Park, PA 16802, USA

*Corresponding author: Department of Evolutionary Anthropology, Duke University, 130 Science Drive, Durham, NC 27708, USA. Email: amy.goldberg@duke.edu

Population genetic theory, and the empirical methods built upon it, often assumes that individuals pair randomly for reproduction. However, natural populations frequently violate this assumption, which may potentially confound genome-wide association studies, selection scans, and demographic inference. Within several recently admixed human populations, empirical genetic studies have reported a correlation in global ancestry proportion between spouses, referred to as ancestry-assortative mating. Here, we use forward genomic simulations to link correlations in global ancestry proportion between mates to the underlying mechanistic mate choice process. We consider the impacts of 2 types of mate choice model, using either ancestry-based preferences or social groups as the basis for mate pairing. We find that multiple mate choice models can produce the same correlations in global ancestry proportion between spouses; however, we also highlight alternative analytic approaches and circumstances in which these models may be distinguished. With this work, we seek to highlight potential pitfalls when interpreting correlations in empirical data as evidence for a particular model of human mating practices and to offer suggestions toward development of new best practices for analysis of human ancestry-assortative mating.

Keywords: assortative mating; human admixture; genetic ancestry; global ancestry proportion; local ancestry tracts; forward genetic simulation; individual-based simulations; SLiM

## Introduction

Nonrandom mating has long been appreciated as an important source of genetic structure in natural populations (Fisher 1918; Wright 1921, 1950; Nagylaki 1978). Positive assortative mating (hereafter, assortative mating), wherein genotypic or phenotypic trait values are positively correlated between mates, has been empirically observed across animal species (Jiang et al. 2013). Theoretical and empirical studies have demonstrated the consequences of assortative mating for speciation and hybridization (Kondrashov 1983; Otto et al. 2008; Tung et al. 2012; Schumer et al. 2017; Kopp et al. 2018; Powell et al. 2021; Muralidhar et al. 2022; Natola et al. 2022; Smadja et al. 2022; Robinson et al. 2023) and for the distributions of traits within populations (Wright 1921; Norris et al. 2019; Kim et al. 2021; Border et al. 2022; Muralidhar et al. 2022; Horwitz et al. 2023).

In nonhuman primates that live in complex and structured social environments, sociodemographic factors have also been demonstrated to influence mate pair formation (Keddy-Hector 1992; Klinkova et al. 2005; Setchell and Wickings 2006; Van Belle et al. 2009; Tung et al. 2012; Fogel et al. 2021). As such, when it comes to humans, assortative mating has been the purview not only of biologists but also of social scientists (e.g. Buss and Barnes 1986; Mare 1991; Kalmijn 1998; Luo and Klohnen 2005; Blossfeld 2009; Torche 2010; Schwartz 2013; Greenwood et al. 2014; Henz and Mills 2017; Smieja and Stolarski 2018; Chiappori 2020; De La Mare and Lee 2023). Positive correlations have been reported between human spouses for a diverse array of phenotypes, including morphometric measurements, health outcomes, personality traits, lifestyle factors, age, socioeconomic status, educational attainment, religious affiliation, and language (Nagoshi et al. 1990; Robinson et al. 2017; Fibla et al. 2022; Horwitz et al. 2023; Yamamoto et al. 2023). These associations are likely driven by multiple generative processes, including individual mate choice preferences, phenotypic convergence over time facilitated by cohabitation, and social structures that restrict or promote particular pairings. These processes may act individually or in concert. For instance, sociological literature on assortative mating by educational attainment has revealed the influence of both a preference for mate similarity (Kalmijn 1998) and social barriers to marriage across socioeconomic class (Torche 2010) and that the strength of assortment by education over time is sensitive to the degree of temporal overlap between the end of schooling and average age at the time of marriage (Mare 1991), suggesting that social milieu is likely also an important factor in structuring mate pair outcomes.

Within the field of population genetics, relatively less attention has been paid to understanding the contributions of these generative processes to producing the observed correlations between mating pairs. Indeed, mechanism and outcome are often conflated or not clearly delineated when discussing assortative mating, which obscures these multiple mechanisms that may contribute to mate similarity. When studies do consider mechanism, they typically draw on foundational work in the sexual selection literature modeling the effects of female choice on male

phenotype (Lande 1981; Kirkpatrick 1982; Seger 1985), in which the mechanism driving assortative mating is assumed to be mate choice. To extend this framework beyond sexual dimorphism, the mate choice mechanism is modeled as a preference for mates that "match" an individual's own phenotype (Kopp *et al.* 2018; Goldberg *et al.* 2020; Kim *et al.* 2021; Muralidhar *et al.* 2022). However, some studies have suggested that temporal structure in the mating process (Xie *et al.* 2015; Woodman *et al.* 2023) or non-uniform ability to attract mates (Burley 1983) can generate phenotypic correlations between mates without an explicit preference for phenotypic similarity. In this paper, we use the term "assortative mating" to refer specifically to an empirical observation of greater resemblance between mates than expected by chance, agnostic to mechanism.

Reports of assortative mating by ancestry in humans provide a particularly intriguing example for exploring mechanism because genetic ancestry is a complex quantitative "phenotype" that involves every locus in the genome and that can only be ascertained by sequencing. In recently admixed human populations, in which individuals derive ancestry from multiple source populations, a positive correlation in global ancestry proportion between spouses has been observed. This phenomenon, referred to as ancestry-assortative mating, has been reported in multiple Latino populations (Risch *et al.* 2009; Zou *et al.* 2015; Spear *et al.* 2020; Mas Sandoval *et al.* 2023), as well as in African-Americans (Zaitlen *et al.* 2017; Avadhanam and Williams 2022), Cabo Verdeans (Korunes *et al.* 2022), and Ni-Vanuatu (Arauna *et al.* 2022). Ancestry-associated mating patterns, inferred from excess sharing of single-nucleotide variants, have also been reported for non-Hispanic white spouse pairs in the United States (Sebro *et al.* 2010, 2017; Domingue *et al.* 2014). Empirical studies commonly use such correlations in global ancestry proportion between spouses to suggest that ancestry shapes mate choice, although this interaction would necessarily be mediated by a proxy phenotype of some type not by quantitative ancestry proportion. However, the relative contributions of individual preferences and broader social mechanisms to generating these correlations are unclear.

Accounting for assortative mating is critical for accurate statistical and population genetic analysis in admixed human populations: it has been shown that assortative mating confounds genome-wide association studies (Howe *et al.* 2021; Border *et al.* 2022; Veller and Coop 2024) and heritability estimation (Sebro and Risch 2012; Tenesa *et al.* 2016; Huang *et al.* 2024). In addition, ancestry-assortative mating specifically has been shown to produce estimates for the timing of admixture that are too recent (Zaitlen *et al.* 2017; Goldberg *et al.* 2020; Korunes *et al.* 2022). However, it remains challenging to establish a null expectation in the presence of assortative mating without understanding the underlying mechanism at play.

To date, statistical methods to correct for ancestry-assortative mating in empirical data have tended to generate null models by simulating data to match the observed empirical correlation in global ancestry proportion between mating pairs, often without a specific mechanistic model (Zaitlen *et al.* 2017; Spear *et al.* 2020; Pfennig and Lachance 2023; Huang *et al.* 2024). This approach makes 2 important assumptions: first, the correlation coefficient for a given population has remained constant over time and second, matching the correlation coefficient is sufficient to recapitulate the dynamics of global and local ancestry. In contrast, theoretical population genetic studies have examined the behavior of these same parameters using mechanistic models of biased mate choice (Goldberg *et al.* 2020; Kim *et al.* 2021; Muralidhar *et al.* 2022). This approach assumes that ancestry-based preference is

the primary mechanism driving nonrandom mating. These studies typically do not directly consider whether the model used results in an observed correlation in global ancestry proportion between mates, which leaves open the question of how applicable their conclusions are to the context of empirical studies of human ancestry-assortative mating.

Here, we used forward-in-time simulations to probe the relationship between mate choice mechanism and observed correlation in global ancestry proportion, comparing variants of 2 classes of mate choice model. The first model class considers only individual mate choice preferences for similarity in global ancestry proportion, treating genetic ancestry as a continuous quantitative trait. The second class considers only discrete social groups, assigning each individual to 1 of 2 groups and limiting cross-group mating events. (Importantly, group membership is "inherited" from one parent, and group identities are initially associated with source populations; see *Methods*.) We found that ancestry-similarity and social group models can produce similar correlations in global ancestry proportion between mates, suggesting that either or both mechanisms could be relevant in interpreting empirical data. Furthermore, the correlation coefficient was not constant over time in any of our simulations, and not all variants of the ancestry-similarity class or all parameter values of the social group model maintained sufficient population variance to sustain correlations in global ancestry proportion over tens of generations. Our results highlight important caveats about the assumptions behind both existing approaches and suggest additional analyses of empirical data that may help reveal the underlying mechanism of ancestry-assortative mating in human populations.

## Methods
### Simulation framework

We performed forward-in-time simulations of admixture with equal contributions from 2 source populations using SLiM 4.0.1 (Haller *et al.* 2019; Haller and Messer 2023). We modeled sexually reproducing diploid individuals (census population size $N = 10,000$/generation) with 22 independently segregating chromosomes, representing the size distribution of human autosomes (total genome size $L = 2.88$ Gb), with a uniform recombination rate $r = 1 \times 10^{-8}$. Unless otherwise noted, there was a single pulse of admixture and no additional contributions from the source populations were introduced at later generations. We did not model any type of 1D or 2D spatial relationships between individuals, nor did we simulate genotypes.

For each of 50 generations post-admixture, we extracted the position and source of local ancestry tracts from the tree sequence (Haller *et al.* 2019). From these data, we calculated the total proportion of the genome that each individual derived from source population 1 (hereafter, global ancestry proportion, $x \in [0, 1]$). Separately, we tracked the social group $s \in \{A, B\}$ to which each individual belonged: offsprings were assigned to the same social group as their first parent—i.e. without reference to their global ancestry proportion. We focus primarily on results from the first 20 generations post-admixture, approximately corresponding to the timing of African-European admixture initiated by the trans-Atlantic slave trade (Zaitlen *et al.* 2017; Hamid *et al.* 2021; Korunes *et al.* 2022; Mas Sandoval *et al.* 2023; Mooney *et al.* 2023), with a subset of results over longer time periods in the supplementary material.

We considered 2 broad classes of mate choice models: (1) *ancestry-similarity*, wherein the probability of an individual with

global ancestry proportion $x_i$ mating with an individual with global ancestry proportion $x_j$ is defined in terms of $|x_i - x_j|$ and (2) *social group*, wherein the probability of an individual belonging to social group $s_i$ mating with an individual belonging to social group $s_j$ is defined by whether or not $s_i = s_j$.

To implement nonrandom mating in SLiM, we started from the default Wright–Fisher framework, which is designed to allow females to be selective if a `mateChoice()` callback is used. In this work, we did not distinguish males and females, instead allowing each individual to potentially serve as both parent 1 (selector) and parent 2 (selected) in sequential mating events. For each mating event, parent 1 was uniformly sampled with replacement from the population, while parent 2 was sampled proportional to mating weight $\psi_{i,j}$, calculated according to the specified mate choice function (defined below) evaluated for $f(x_i, x_j)$ or $f(s_i, s_j)$, under the ancestry-similarity and social group models, respectively (Fig. 1). Incidental selfing was explicitly prohibited, and each mating event generated a single child.

## Defining a common parameter for model comparison

The mate choice functions we considered (Equations 2–5, below) differ in how they parameterize the strength of mating bias; to aid in comparison across models, we will refer throughout to $\alpha \in [0, \infty)$, the frequency of mating events between individuals from opposite source populations. Specifically, we define $\alpha$ to be inversely related to the proportion of admixed offspring in the initial generation post-contact, A, assuming that the 2 source populations admix in equal proportions. We have:

$$\alpha = \frac{1}{A} - 1. \tag{1}$$

That is, at time of population contact, individuals are $\alpha$ times more likely to choose a mate from within their own source population than from the opposite source population. Increasing values of $\alpha$ represents an increasing bias toward endogamy, with $\alpha = 0$ corresponding to exclusive exogamy, $\alpha \in (0, 1)$ to negative assortative mating, $\alpha = 1$ to random mating, and $\alpha > 1$ to positive assortative mating. Given our focus on mechanisms that might explain empirically observed positive correlations in global ancestry proportion between mates, we consider only $\alpha \geq 1$. While our models differ in their dynamics over time, simulations with the same $\alpha$ have the same correlation in global ancestry proportion between mates, $r(x_i, x_j) \in [0, 1]$, for the first generation post-admixture under all models.

## Ancestry-similarity models

Individual-based mate preference models originate in the sexual selection literature, where they were developed for studying the coevolutionary dynamics of male secondary sex characteristics and female mate choice (Lande 1981; Kirkpatrick 1982; Seger 1985). Generally, under these models, male phenotype and female preference are controlled by distinct genetic loci (or sets of loci). In the speciation literature, an alternative family of models is concerned with assortative mating as a mechanism of sympatric speciation and model mate preference based on phenotypic similarity between mates for some ecologically relevant trait (Dieckmann and Doebeli 1999; Burger and Schneider 2006; Pennings et al. 2008; Rettelbach et al. 2013). These "matching rule" models (Kopp et al. 2018) are the basis for our ancestry-similarity model class, wherein individuals preferentially choose mates to maximize similarity in phenotype (*i.e.* global ancestry proportion). Importantly for the models we consider here, these are fixed relative preference models: a focal

individual $i$ assigns each potential mate a preference value $\psi_{i,j}$ that depends only on the potential mate's phenotype and is density independent (Seger 1985).

Ancestry-similarity models define a mate choice function $\psi_{i,j} = f(x_i, x_j)$ for calculating the sampling weight assigned to individual $j$ with global ancestry proportion $x_j$ as a potential mate for a focal individual $i$ with global ancestry proportion $x_i$. In the present study, we considered 3 variants of the ancestry-similarity model identified in a literature search, which differ in the precise definition of the mate choice function $f$.

Models of ancestry-based assortative mating typically model $f(x_1, x_i)$ as an exponential function, such that $\psi_{i,j}$ decays exponentially at rate $c = \ln(\alpha)$ as $|x_i - x_j|$ increases. We define:

$$\psi_{i,j} = e^{-c|x_i - x_j|}. \tag{2}$$

We refer to this as the *stationary preference model* ["like-with-like" model in (Muralidhar et al. 2022)]: for a given absolute difference in global ancestry proportion, $|x_i - x_j|$, the associated preference $\psi_{i,j}$ does not change over time. It is important to note, however, that the range of $\psi_{i,j}$ is constrained by $x_i$: an individual $i$ with global ancestry proportion $x_i = 0.5$ is more similar to all potential mates than an individual with $x_i = 0$ and is therefore less selective in choosing a mate (Fig. 1a). Furthermore, without additional migration from the source populations, variance in global ancestry proportion across individuals decreases over time (Supplementary Fig. 2), meaning that the pool of potential mates is becoming increasingly homogenous and the ratio of $\max(\psi_{i,j})/\min(\psi_{i,j})$ approaches 1 for all values of $x_i$ (Supplementary Fig. 1a). In other words, all individuals become less selective in choosing mates over successive generations.

One approach to account for the decreased variance in global ancestry proportion over time, and thus preserve mate selectiveness, is to decrease $c$, rather that holding it constant. Following (Kim et al. 2021), we rescale $c$ in each generation $t$ using the variance in global ancestry proportion observed in that generation, $\sigma_x^2(t)$:

$$\psi_{i,j} = e^{\frac{-c}{\sigma_x^2(t)} |x_i - x_j|}. \tag{3}$$

We refer to this as the *increasing preference model* [equation A2 in the study by Kim et al. (2021)]. Under this model, as under the stationary preference model, an individual $i$ with global ancestry proportion $x_i = 0.5$ is less selective than one with $x_i = 0$ (Fig. 1b). However, because $\sigma_x^2(t)$ decreases over time, there is an increase in mate selectiveness over successive generations under this model for a given $x_i$ (Supplementary Figs. 1b and 3).

We also considered a third ancestry-similarity model, employing a Gaussian—rather than exponential—mate choice function. This approach is commonly used in models of assortative mating based on a quantitative trait (e.g. Dieckmann and Doebeli 1999; Pennings et al. 2008; Funk et al. 2021), although it has not to our knowledge been used to model assortative mating based on global ancestry proportion. Under this model, $\psi_{i,j} \sim N(x_i, \sigma^2)$ where $x_i$ is the global ancestry proportion of the focal individual and the constant $\sigma^2 = \frac{1}{\sqrt{-2 \ln(\alpha^{-1})}}$ determines the strength of mate choice preference. Again, $\psi_{i,j}$ decreases as the difference in global ancestry proportion between potential mates increases:

$$\psi_{i,j} = \frac{1}{\sigma\sqrt{2\pi}} e^{-\frac{(x_i - x_j)^2}{2\sigma^2}}. \tag{4}$$

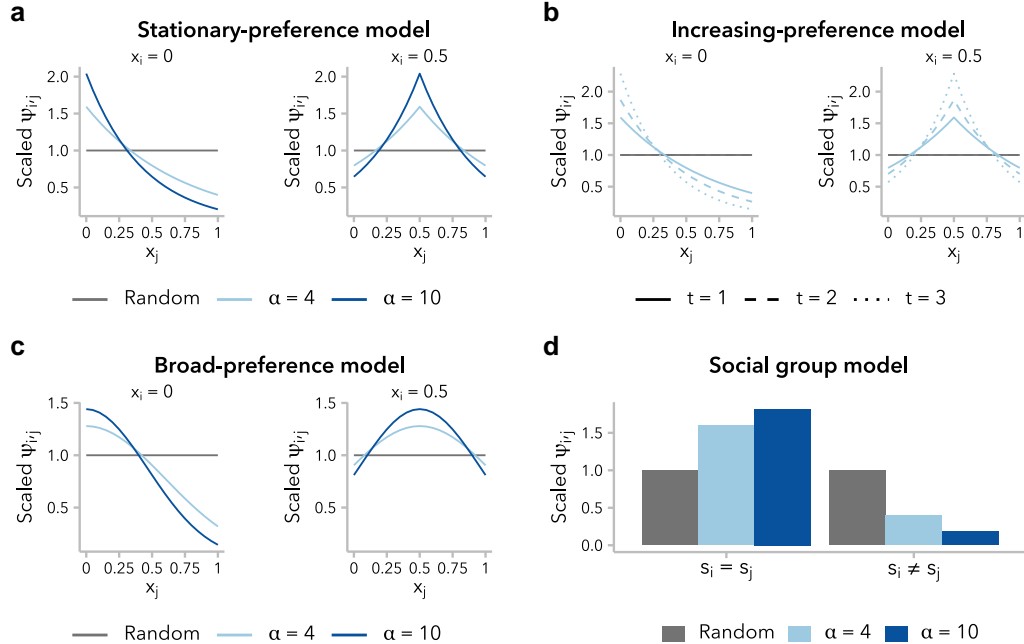

**Fig. 1.** Four functions for defining the sampling weight, $\psi_{i,j}$, assigned to individual $j$ as a potential mate for a focal individual $i$, based on global ancestry proportion $x_j$ a–c) or social group $s_j$ d). For each mating event, mate 1 was uniformly sampled from the population, and mate 2 was sampled proportional to $\psi_{i,j}$. For the sake of visualization, values of $\psi_{i,j}$ have been scaled such that $\psi_{i,j} = 1$ means that the probability of individual $j$ being selected as a potential mate for individual $i$ is $1/n$ ($n$, number of potential mates). All mate choice functions are reparameterized in terms of $\alpha$. a) The stationary preference variant of the ancestry-similarity model [Equation (2)] defines $\psi_{i,j}$ as a function of the difference in global ancestry proportion between individuals, $|x_i - x_j|$. b) Under the increasing preference variant of the mate preference model [Equation (3)], the value of $\psi_{i,j}$ is scaled to the variance in global ancestry proportion, $\sigma_x^2(t)$, across all potential mates, resulting in increasing choosiness over time. $\psi_{i,j}$ values are shown for $t \in \{1, 2, 3\}$ and $\alpha = 4$, assuming $\sigma_x^2(t) = 2^{-\frac{t}{2}}$ (Liang *et al.* 2024). c) Under the broad preference variant of the ancestry-similarity model [Equation (4)], $\psi_{i,j}$ follows the probability density function of a normal distribution with mean $x_i$. d) Under the social group model (Equation 5), $\psi_{i,j}$ can take 1 of 2 discrete values, determined by whether a potential mate belongs to the same social group as the focal individual ($s_i = s_j$) or to the other social group ($s_i \neq s_j$).

We refer to this as the *broad preference model* because $f(x_i, x_j)$ decays more gradually with increasing differences in global ancestry proportion, relative to the stationary preference and increasing preference models (Fig. 1c). Thus, a large difference in global ancestry proportion from a potential mate is strongly disfavored, while smaller differences within the pool of "similar" mates are not weighed strongly. Compared with the other ancestry-similarity models, individuals are less selective in choosing mates under the broad preference model; however, as under the stationary preference model, they also become less selective over time (Supplementary Figs. 1c and 4).

## Social group model

An alternative class of models considers preferential mating based on a categorical, rather than quantitative, trait. This type of model is often used in the context of interspecific hybrid zones, wherein nonrandom mating can maintain species boundaries and/or promote hybrid speciation (Melo *et al.* 2009; Schumer *et al.* 2017; Powell *et al.* 2021; Natola *et al.* 2022; Smadja *et al.* 2022). Mechanistically, this can be construed as a model of biased mating by species identity or source population (Goldberg *et al.* 2020). However, application of this type of model in humans is not straightforward: the source populations in question are not biologically distinct species, but rather labels describing socially and geographically defined boundaries that have changed over time. Furthermore, it is unclear how we ought to model the mating behavior of admixed individuals: neither random mating by admixed individuals nor a simple preference of admixed individuals to mate with one another seems likely to produce the observed correlation in global ancestry proportion between mates. Yet, the strongest evidence for ancestry-assortative mating in

humans is observed in contexts where both spouses have admixed ancestry (Risch *et al.* 2009; Zou *et al.* 2015; Zaitlen *et al.* 2017; Spear *et al.* 2020; Avadhanam and Williams 2022; Korunes *et al.* 2022; Mas Sandoval *et al.* 2023).

Social identity is an important organizer of pair formation in humans, as evidenced by widespread cultural practices of endogamy based on race, ethnicity, religion, socioeconomic status, and caste. Although distinct from genetic ancestry, social categorizations often interact with ancestry. For instance, skin pigmentation, a prominent contributor to how an individual is racialized, is correlated with West African-related global ancestry proportion in multiple populations with recent admixture history (Parra *et al.* 2003; Shriver *et al.* 2003; Bonilla *et al.* 2004; Beleza *et al.* 2012). Race has also served as the basis for numerous sociolegal barriers to mating across contexts, including anti-miscegenation laws (Browning 1951). This type of interaction opens the possibility that social barriers to mating might be "recorded" in patterns of ancestry-assortative mating detectable by genomic analyses.

We considered a simple *social group model* with 2 groups, with sampling weight $\psi_{i,j}$ determined by the social group identity of the focal individual and potential mate, $s_i$ and $s_j$, respectively, with preference strength $\alpha$ (Fig. 1d; Supplementary Fig. 5). Rearranging Equation (1):

$$\psi_{i,j} = \begin{cases} 1 - \dfrac{1}{\alpha + 1}, & \left| s_i = s_j \right. \\ \dfrac{1}{\alpha + 1}, & \left| s_i \neq s_j \right. \end{cases} \tag{5}$$

To generate an association between social group membership and global ancestry proportion, we assigned individuals in each source

population to the same social group prior to admixture. However, from the onset of population contact, individuals "inherited" their social group membership from the first parent in the mating pair. Thus, individuals whose parents belonged to opposite social groups were equally likely to be assigned to either social group.

Several features distinguish the social group model from the ancestry-similarity models described above or from one in which individuals use an observable quantitative trait (e.g. skin pigmentation) as a proxy for an unobserved trait (global ancestry proportion). First, social group membership is correlated with—but not determined by—genotype. This matters because it has previously been demonstrated that recombination is expected to decouple the relationship between global ancestry proportion and a proxy quantitative trait over time, leading to only a transient correlation between the 2 phenotypes (Kim *et al.* 2021). Because social group membership is not genetic, it is not impacted by recombination. In addition, individuals are uniformly likely to cross the social group barrier in choosing a mate and do not express relative preferences among potential mates within a social group. Thus, in contrast to an ancestry-similarity model, individuals with intermediate ancestry are not less selective than individuals with more source-like ancestry in choosing a mate.

However, these features of the social group model also mean that the association between global ancestry proportion and social group membership is expected to decay over successive generations, and we explored this relationship quantitatively over time.

## Results

### Not all simulations of nonrandom mating show a positive correlation in global ancestry proportion between mates 20 generations post-admixture

Ancestry-assortative mating is often inferred from a positive correlation in global ancestry proportion between spouses at single point in time and statistical methods to account for nonrandom mating typically generate null models using simulations where potential mate pairs are permuted until the empirically observed correlation is achieved (Zaitlen *et al.* 2017; Pfennig and Lachance 2023; Huang *et al.* 2024). This approach assumes that the correlation observed in the present-day samples has remained constant since the start of admixture. Figure 2 plots the Pearson correlation coefficient for global ancestry proportion between mates, $r(x_i, x_j)$, over time under 3 variants of the ancestry-similarity model or the social group model, for multiple strengths of mating bias, $\alpha$, and following a single pulse of admixture. In all scenarios of nonrandom mating that we considered, we observed positive correlations in global ancestry proportion between mates (Fig. 2; Supplementary Fig. 6). However, contrary to the assumption that the correlation observed in a contemporary sample should be modeled as a fixed value, the observed correlation coefficients in our simulations decayed over time. In simulations performed under the increasing preference model, $r(x_i, x_j)$, reached a stable plateau within the first 10 generations post-admixture across values of $\alpha$ (Supplementary Fig. 7a). In contrast, simulations performed under the other 2 variants of the ancestry-similarity model (stationary preference or broad preference) approached $r(x_i, x_j) = 0$ within 20 generations post-admixture, unless $\alpha$ was sufficiently large to disrupt the admixture process altogether (Supplementary Figs. 8 and 9). Under the social group model, positive values of $r(x_i, x_j)$ could be observed 20 generations post-admixture for $\alpha > 2$, although these models also approached $r(x_i, x_j) = 0$ on longer time scales (Supplementary Fig. 7b).

To formally test whether the decay in $r(x_i, x_j)$ meant that mating was effectively random at $t = 20$ generations postadmixture under the stationary preference and broad preference models, we permuted the mating pairs and recalculated $r(x_i, x_j)$ for 1,000 permutations of the simulated data. For the stationary preference model, we found examples in which $r(x_i, x_j)$ was close to—but still significantly different from—zero by a permutation test. For instance, for the 5 replicate simulations with $\alpha = 10$, $r(x_i, x_j) \in$ [0.0182, 0.0444] for $t = 20$ generations post-admixture. In 2 of these simulations, the observed $r(x_i, x_j) \in \{0.0399, 0.0444\}$ never overlapped the permuted distribution (empirical $P < 0.001$). In contrast, $r(x_i, x_j)$ always overlapped the permuted distribution after $t = 11$ generations under the broad preference model ($\alpha = 10$) (Supplementary Fig. 10).

Thus, not all variants of the ancestry-similarity model generate a positive correlation in global ancestry proportion between mates that is observable 20 generations post-admixture while also allowing for a large population of individuals with intermediate global ancestry proportion. Conversely, results for the social group model demonstrate that even relatively weak social barriers can produce a positive correlation in global ancestry proportion between mates on the timescale of tens of generations, even as the association between social group membership and global ancestry proportion diminishes over time (Supplementary Fig. 11).

### Relationship between correlation in global ancestry proportion between mates and variance in global ancestry proportion across individuals differs between models

To understand why the stationary preference and broad preference models did not produce a sustained correlation in global ancestry proportion between mates, $r(x_i, x_j) \gg 0$, over time, while the increasing preference and social group models sometimes did (Fig. 2), we next considered the variance in global ancestry proportion across individuals in generation $t$, $\sigma_x^2(t)$. Intuitively, nonrandom mating requires that individuals in the population vary for the trait that serves as the basis for mate choice: if all potential mates are identical for the relevant phenotype, mating pairs will form randomly regardless of the strength of mate choice bias. Thus, we hypothesized that our simulations of biased mating would become indistinguishable from random mating when $\sigma_x^2(t)$ became sufficiently small.

Consistent with prior analytic models, $\sigma_x^2(t)$ was initially greater under nonrandom relative to random mating, regardless of mate choice model (Wright 1921; Verdu and Rosenberg 2011; Zaitlen *et al.* 2017; Goldberg *et al.* 2020; Liang *et al.* 2024) and decayed more slowly for larger $\alpha$ (Supplementary Fig. 12). However, for simulations under the stationary preference and broad preference models, $\sigma_x^2(t)$ approached zero within approximately $t = 15$ generations post-admixture under our set of parameters. This timing corresponds to when $r(x_i, x_j) \approx 0$ under these 2 models, as observed in Fig. 2. In contrast, under the increasing preference and social group models, which maintained a positive $r(x_i, x_j)$ at $t = 20$ generations, $\sigma_x^2(t)$ was greater than for random mating for any generation $t$ (all $\alpha$ for the increasing preference model; $\alpha > 2$ for the social group model).

As predicted, higher $\sigma_x^2(t)$ corresponded to higher $r(x_i, x_j)$ for all 4 models, at least for early generations post-admixture (Supplementary Fig. 13). However, this relationship was not linear and differed across both models and ranges of $\sigma_x^2(t)$. In early generations, when $\sigma_x^2(t)$ is large, $r(x_i, x_j)$ scales approximately linearly with $\sigma_x^2(t)$ under all models (Fig. 3a). As the admixture process continues, however, we observed that for intermediate values of $\sigma_x^2(t)$,

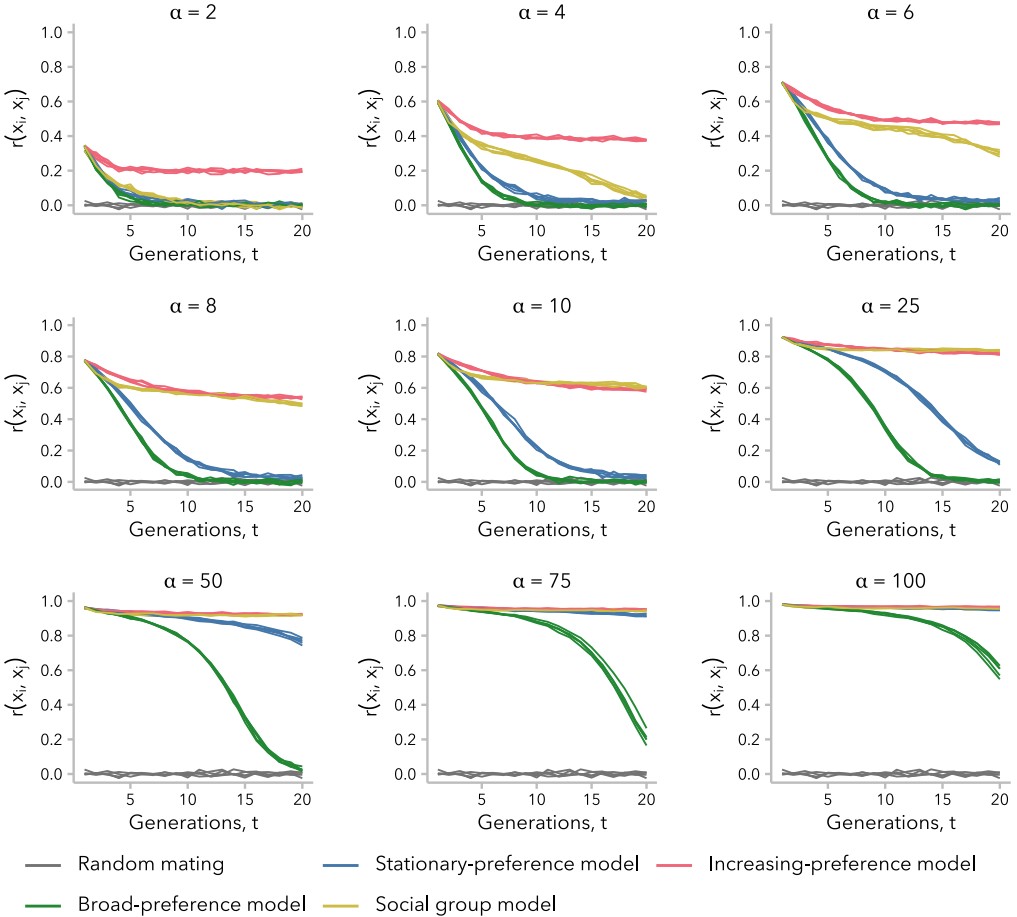

**Fig. 2.** Correlation in global ancestry proportion between mates, $r(x_i, x_j)$, was not constant over time and decayed to near-zero within 20 generations post-admixture in some simulations. Simulations under the stationary preference [Equation (2)] and broad preference [Equation (4)] models did not simultaneously maintain $r(x_i, x_j) \gg 0$ while also allowing for a large population with intermediate global ancestry proportion values, whereas those under the increasing preference [Equation (3)] and social group [Equation (5)] models did for some $\alpha$. By definition, $r(x_i, x_j)$ in the first generation after admixture is the same across all simulations performed with the same value of $\alpha$, regardless of mate choice model. For the sake of comparison, the same replicate random mating simulations ($\alpha = 1$) are reproduced in each subplot. See Supplementary Fig. 6.

the same $\sigma_x^2(t)$ corresponded to a lower $r(x_i, x_j)$ under the stationary preference and broad preference models relative to the other 2 models (Fig. 3, b and c). In other words, simulations under these 2 models required greater variance in global ancestry proportion across individuals in order to observe the same correlation in global ancestry proportion between mates. By $t = 20$ generations post-admixture, $\sigma_x^2(t)$ is small, and we observed that the same $\sigma_x^2(t)$ corresponded to a higher $r(x_i, x_j)$ under the increasing preference model than under the social group model (Fig. 3d). This emphasizes a key distinction between these 2 models: the social group model sustains a positive $r(x_i, x_j)$ over tens of generations by maintaining higher $\sigma_x^2(t)$, whereas the increasing preference model instead compensates for very low $\sigma_x^2(t)$ by scaling the mate choice parameter $c$ by $\sigma_x^2(t)$, increasing the strength of preference in each generation.

Given our broader objective of understanding mechanisms that could potentially explain the correlation in global ancestry proportion between spouses that has been reported in empirical data, in the following sections we focus on the 2 models that produce $r(x_i, x_j) \gg 0$ for tens of generations post-admixture: the increasing preference model and the social group model. Specifically, we compare pairs of simulations performed under these models that produced the same value of $r(x_i, x_j)$ at $t = 20$ generations post-admixture, noting that we did not observe evidence

for any effect of global ancestry proportion on reproductive success in these simulations (Supplementary Figs. 14 and 15).

## The same correlation in global ancestry proportion between mates can be explained by multiple patterns of nonrandom mating

Though commonly used for empirical genetic studies, the correlation in global ancestry proportion between mates is not particularly informative about the underlying distribution of global ancestry proportion, $x$, within the population. As a general principle, the admixture process ultimately results in a shift in the distribution of $x$ over time from bimodal to unimodal, as 2 discrete source populations with $x \in \{0, 1\}$ converge to a continuous distribution of intermediate $x$ values. This shift to a unimodal distribution occurs within a single generation under random mating; under ancestry-biased mating, we expect the shift to be delayed, as individuals with more extreme $x$ preferentially mate with one another.

We observed that the timing of this shift is delayed even more under the social group model than under the increasing preference model, whether comparing pairs of simulations matched for mating bias strength, $\alpha$ (Supplementary Fig. 16) or for correlation in the global ancestry proportion, $r(x_i, x_j)$. Specifically, we considered the distributions of $x$ at $t = 20$ generations post-

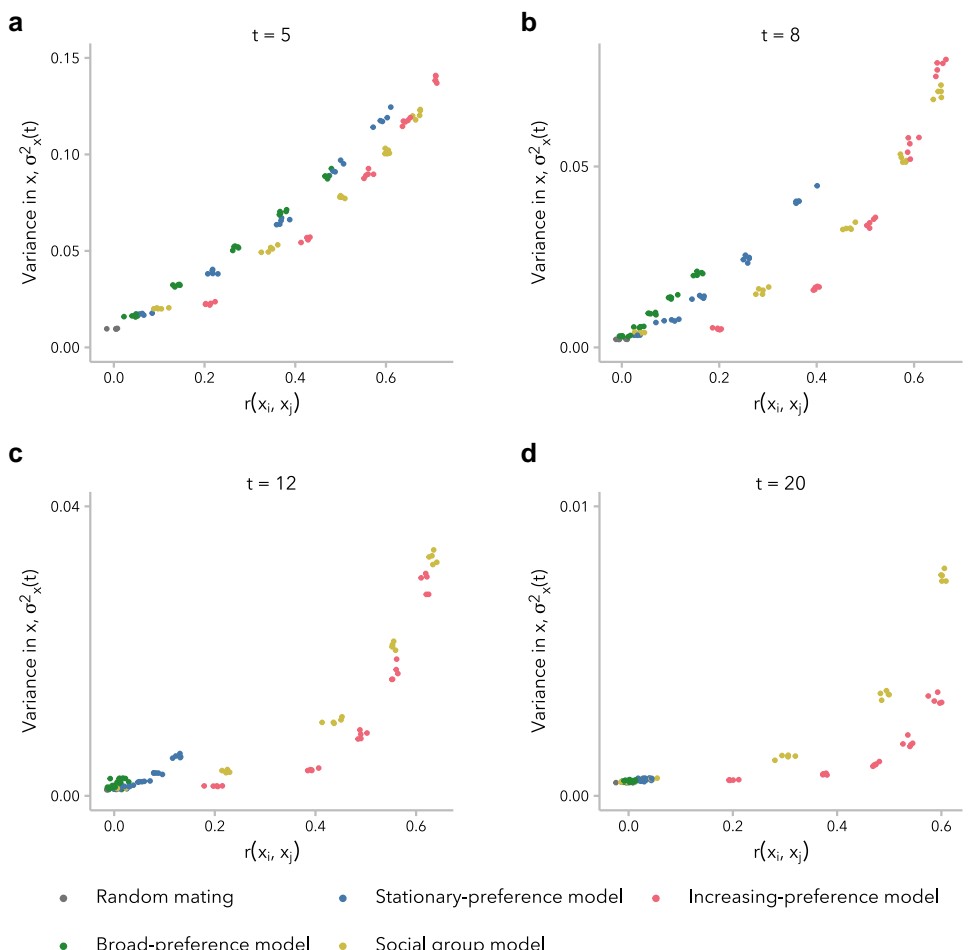

**Fig. 3.** The relationship between variance in global ancestry proportion across individuals, $\sigma_x^2(t)$, and correlation in global ancestry between mates, $r(x_i, x_j)$, differs between models and changes as $\sigma_x^2(t)$ approaches zero. a) In early generations, when $\sigma_x^2(t)$ is larger, there is a quasi-linear relationship between $r(x_i, x_j)$ and $\sigma_x^2(t)$. b and c) As the value of $\sigma_x^2(t)$ decreases, differences appear between the 2 models that sustain high $r(x_i, x_j)$ and the 2 model that do not. d) At generation $t = 20$, a given value of $r(x_i, x_j)$ corresponds to a lower $\sigma_x^2(t)$ under the increasing preference model relative to the social group model. In other words, the increasing preference model maintains a high $r(x_i, x_j)$ at $t = 20$ generations post-admixture despite low differentiability of potential mates. Each dot represents 1 simulation (5 replicates each for $\alpha \in \{1, 2, 4, 6, 8, 10\}$).

admixture, approximating the timing of the onset of admixture for many human populations with African and European ancestry (Zaitlen *et al.* 2017; Hamid *et al.* 2021; Korunes *et al.* 2022; Mas Sandoval *et al.* 2023; Mooney *et al.* 2023). At $t = 20$, we found a unimodal distribution of ancestry proportion for simulations under the increasing preference model for all $\alpha$, whereas the social group model produced a bimodal distribution for simulations with $\alpha \geq 7$. For simulations under the social group model with $\alpha < 7$, ancestry proportion was unimodal but with higher variance than simulations under the increasing preference model with the same correlation between mates (Supplementary Fig. 17).

Differences in the distribution of global ancestry proportion, $x$, within the population between the increasing preference and social group models are also reflected in the distribution of mating pairs, $(x_i, x_j)$ at $t = 20$ generations post-admixture. Under the increasing preference model, the mating pairs form a single cluster with highest density around (0.5, 0.5), reflecting the original equal contributions from the 2 source populations (Fig. 4a). In contrast, for a simulation under the social group model with the same correlation in global ancestry proportion between mates, $r(x_i, x_j)$, there were 2 distinct clusters of mating pairs, reflecting preferential mating within each of 2 subpopulations (Fig. 4b).

These clusters were identifiable by eye when $r(x_i, x_j) \geq 0.4$ (Supplementary Fig. 18). Thus, although the same correlation in global ancestry proportion between mates can be produced by both the increasing preference and social group models, the underlying structure of nonrandom mating is not equivalent and may be distinguishable when $r(x_i, x_j)$ is large.

Given that evidence for ancestry-assortative mating is often assessed in substantially smaller empirical datasets, we also examined whether these differences in mating structure could be observed in a subsample ($n = 100$ individuals) of our simulated population. The correlation in global ancestry proportion between mates is typically visualized as a dot-plot (e.g. Zou *et al.* 2015; Korunes *et al.* 2022); we found that the 2 models looked similar when taking this approach (Supplementary Fig. 19). However, even with only 100 individuals, hexagonal bin plots (Carr *et al.* 1987) suggested the presence of 2 discrete clusters of mating pairs in simulations under the social group model with high $\alpha$ [and thus high $r(x_i, x_j)$] (Fig. 4, c and d; Supplementary Fig. 20).

These results, along with similar results using the distribution of the absolute difference in global ancestry proportion between mate pairs, $\Delta_x = |x_i - x_j|$, rather than summarizing it as a correlation coefficient (Supplementary Fig. 21), underscore that

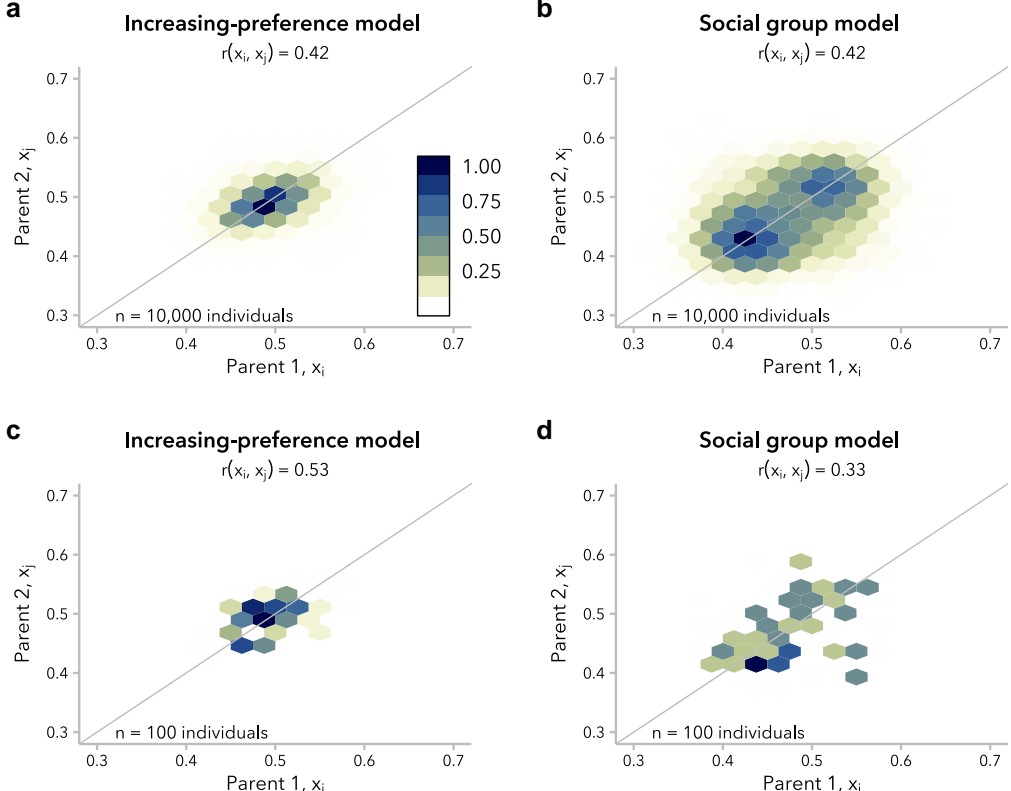

**Fig. 4.** Simulations under the increasing preference ($\alpha = 5$) and social group ($\alpha = 7$) models produced the same correlation in global ancestry proportion between mates, $r(x_i, x_j)$, at $t = 20$ generations post-admixture but with different underlying mating structure. Hexagonal bin plots represent the correlation in global ancestry proportion between the 2 parents of individuals in generation $t = 20$. Each hexagon corresponds to a bin of 0.025 global ancestry proportion units, with color encoding the scaled density (max = 1) of mating pairs in each bin. a) Under the increasing preference model, mating pairs cluster around a single bin of maximum density. b) In contrast, under the social group model, mating pairs form 2 clusters, representing preferential mating within each of 2 social groups. All 10,000 individuals are shown. A $y = x$ line is shown for reference. c and d) The same trends are observed when considering a subsample of $n = 100$ individuals. See Supplementary Figs. 18–20.

multiple mating structures can produce very similar correlation in global ancestry proportion between mates and highlight the utility of alternative visualizations for understanding the mating structure in a population of interest.

## Nonrandom mating produced an excess of long local ancestry tracts relative to random mating, leading to underestimates for the timing of admixture

Thus far, we have focused on the relationship between mate choice model and the distribution of global ancestry proportion. We next considered the impact of model choice on the length distribution of local ancestry tracts, which is used directly or indirectly (e.g. using admixture linkage disequilibrium as a proxy) to infer demographic parameters, including the time since the onset of admixture (Moorjani *et al.* 2011; Gravel 2012; Loh *et al.* 2013; Hellenthal *et al.* 2014). Prior work has shown that nonrandom mating disrupts the decay in local ancestry tract length due to recombination, resulting in long local ancestry tracts consistent with more recent admixture than truly occurred (Zaitlen *et al.* 2017; Korunes *et al.* 2022). Thus, some methods to correct for the systematic underestimation of the time since admixture due to assortative mating attempt to recapitulate local ancestry tract length dynamics by matching the empirical correlation between spouses (Zaitlen *et al.* 2017).

We first compared the median local ancestry tract length at $t = 20$ generations post-admixture in simulations under the increasing preference and social group models, matched for correlation in global ancestry proportion between mates, $r(x_i, x_j)$. Differences in median local ancestry tract length were modest when comparing across $\alpha$ values within a single model, between these 2 mate choice models, and between both models and random mating (Fig. 5a). However, we did observe across all 4 mate choice models that larger values of $\alpha$ were associated with longer median local ancestry tract lengths at generation $t = 20$ (Supplementary Fig. 22). Furthermore, under the increasing preference and social group models—the 2 models under which $r(x_i, x_j) \gg 0$ at generation $t = 20$—greater $r(x_i, x_j)$ was also associated with longer median local ancestry tract length. Additionally, for a given value of $r(x_i, x_j)$, median local ancestry tract length tended to be longest under the social group model (Fig. 5b).

To directly test the effects of the increasing preference and social group models on downstream inference of admixture timing, we fit an exponential to the local ancestry tract length distribution, with decay rate $\lambda = (t + 1) \times m$, where $t$ represents generations postadmixture and $m$ represents the proportion of individuals from source population 1 contributing to the founding of the admixed populations. From the fit, we then inferred that a single-pulse admixture event occurred $t$ generations before the time of sampling [equation 1 from the study by Gravel (2012)]. We observed that the simulated distribution of local ancestry tracts had an excess of long local ancestry tracts at $t = 20$ generations post-admixture relative to the exponential fit, particularly for simulations with greater $r(x_i, x_j)$ (Fig. 6, a and b). Although slight, this mismatch in distribution

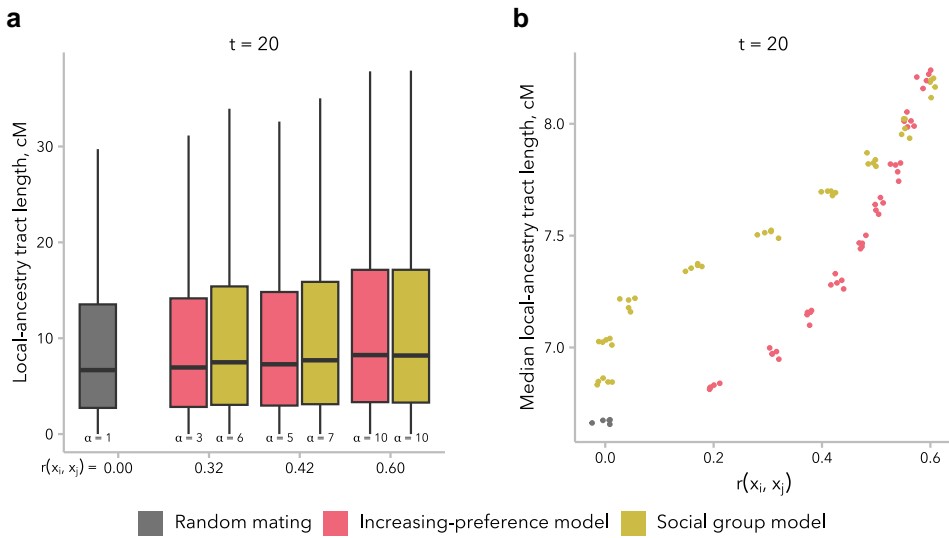

**Fig. 5.** Local ancestry tract length distributions were similar for simulations under the increasing preference and social group models, when comparing simulations with similar correlation in global ancestry proportion between mates, $r(x_i, x_j)$. a) The distribution of local ancestry tract lengths at $t = 20$ generations post-admixture is shown for 3 representative pairs of simulations, matched for $r(x_i, x_j)$. A random mating control simulation is shown for reference. Because the local ancestry tract length distribution has a very long tail, the y-axis is truncated at the top whisker (75th percentile +1.5× the interquartile range). b) Median local ancestry tract length was similar across models and values of $\alpha$. However, for the same $r(x_i, x_j)$, median local ancestry tract length tended to be longer for simulations under the social group model relative to those under the increasing preference model. Each dot represents 1 simulation (5 replicates each for $\alpha \in \{1, \ 2, \ 4, \ 6, \ 8, \ 10\}$).

shape may prove useful as an indication in empirical data that the effects of nonrandom mating should be taken into consideration, although it may also be confused for evidence of continuous migration.

As expected based on prior empirical work (Zaitlen *et al.* 2017; Korunes *et al.* 2022), we underestimated the true time since admixture, and the discrepancy between the truth and our inferred time increased as $r(x_i, x_j)$ increased (Fig. 6c). Intriguingly, we found that this discrepancy was established in the first few generations and then grew relatively slowly over time, suggesting that there might be a plateauing of the effect on longer timescales (Supplementary Fig. 23). For instance, time since admixture was underestimated by an average of 2.25 generations for $t = 20$ and 4.01 generations for $t = 50$ generations post-admixture. As a result, we observe a bias of similar magnitude under all 4 models at $t = 20$ generations post-admixture, although $r(x_i, x_j) \approx 0$ under the stationary preference and broad preference models (Supplementary Fig. 24).

### The effects of continuous migration on the correlation in global ancestry proportion between mates differed between models and values of $\alpha$

Realistic models of human admixture likely include more complex dynamics than a single pulse of admixture followed by complete isolation from the source populations, as we have modeled above. To begin to explore the behavior of the increasing preference and social group models under more complex demographic scenarios, we examined the trajectory of the correlation in global ancestry proportion between mates, $r(x_i, x_j)$, over time in a scenario with continuous migration, wherein 1% of the population in each generation was replaced with migrants from the 2 source populations. Relative to the single-pulse admixture scenario, we might expect $r(x_i, x_j)$ to be smaller in the continuous migration scenario for the same generation $t$ and mating bias strength $\alpha$ because new migrants have fewer potential mates with similar global ancestry proportion to choose from compared to individuals

born in the admixed population. Furthermore, variance in global ancestry proportion across individuals in the admixed population, $\sigma_x^2(t)$, decreases in each successive generation (Supplementary Fig. 12), meaning that new migrants have increasingly dissimilar global ancestry proportion to the average potential mate. On the other hand, mating events between 2 migrants from the same source population will increase $r(x_i, x_j)$.

We found that the balance between these countervailing effects on $r(x_i, x_j)$ differed between models. Under the increasing preference model, $r(x_i, x_j)$ initially decreased over time, similar to the scenario with single-pulse admixture. However, for simulations with $\alpha \geq 4$, there was a subsequent increase over time, resulting in a greater $r(x_i, x_j)$ at $t = 20$ generations post-admixture in the scenario with migration than the one without migration (Fig. 7a). As noted above, the increasing preference model compensates for the decrease in $\sigma_x^2(t)$ by increasing mate selectiveness over time (Supplementary Fig. 12). Consistent with this explanation for the difference between the single-pulse admixture and continuous migration simulations, we observed that the proportion of mating events between migrants increased over time in these simulations (Fig. 7c). Additionally, under the other 2 ancestry-similarity models, which do not compensate for the decreased $\sigma_x^2(t)$ over time, we observed that the prevalence of mating events between migrants was constant and that the value of $r(x_i, x_j)$ was similar with or without migration for a given generation $t$ and $\alpha$ (Supplementary Figs. 25–27).

In contrast, the social group model neither takes into consideration similarity in global ancestry proportion nor increases the strength of mating bias over time. Thus, for simulations under this model, new migrants are equally likely to mate with anyone within their social group regardless of global ancestry proportion; as expected, we observed a constant prevalence in mating events between migrants (Fig. 7d). Additionally, because $\sigma_x^2(t)$ decreases over time, migrants become increasingly dissimilar from potential mates. As a result, continuous migration always decreased $r(x_i, x_j)$

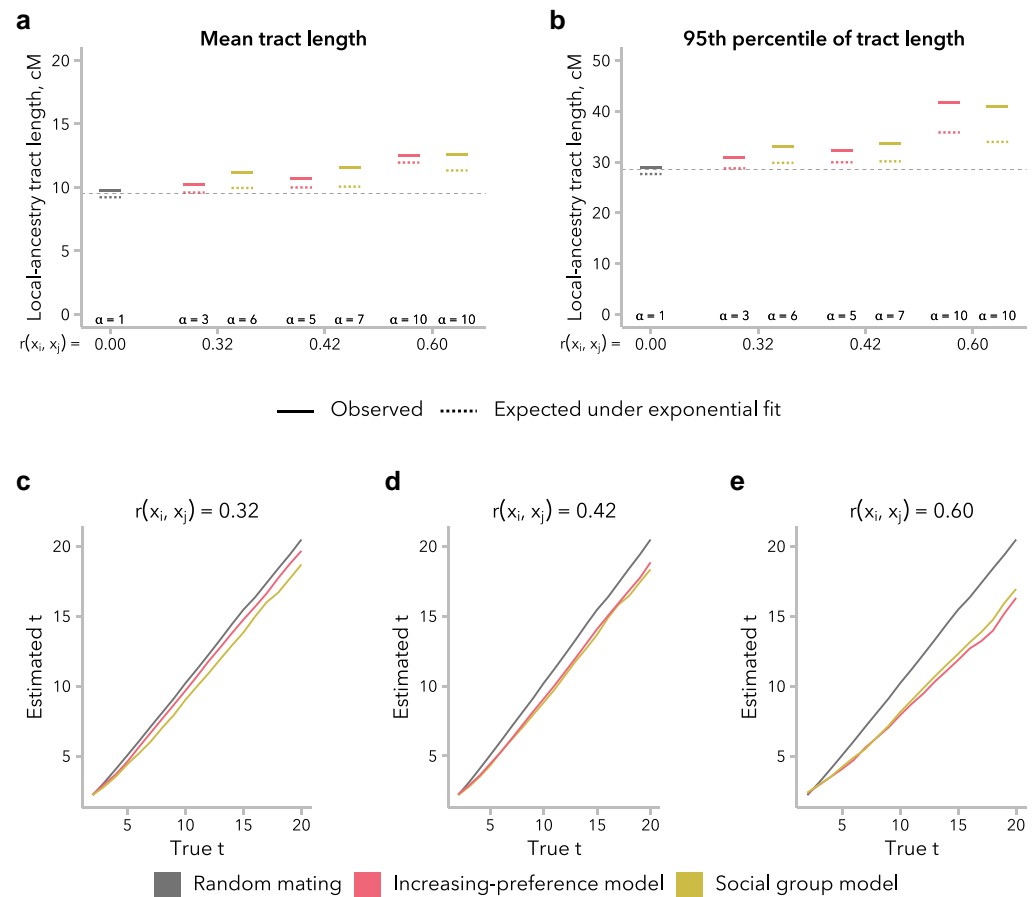

**Fig. 6.** Nonrandom mating led to underestimation of the time since admixture. For each simulation, an exponential model was fit to the local ancestry tract length distribution, with decay rate $\lambda$. a) For the mean local ancestry tract length, observed values (solid lines) were similar to those expected under the exponential fit ($1/\lambda$, dotted lines), indicating a good fit to the data. The dashed line at $y = 9.52$ indicates the expected mean local ancestry tract length for the true values of $t = 20$ and $m = 0.5$. b) However, for the 95th percentile of local ancestry tract length, observed values (solid lines) were larger than those expected under the exponential fit [$-\ln(1 - 0.95)/\lambda$, dotted lines], particularly when $r(x_i, x_j)$ is high. The dashed line at $y = 28.53$ indicates the expected 95th percentile of local ancestry tract length for the true values of $t = 20$ and $m = 0.5$. c–e) Estimated time since admixture was similar between the increasing preference and social group models. As expected, inference was accurate under random mating and underestimated under both models of nonrandom mating, with increasing discrepancy between true and inferred values for larger $r(x_i, x_j)$.

under this model relative to the single-pulse admixture scenario, controlling for generation and $\alpha$ (Fig. 7c).

## Discussion

While it is appreciated that humans, like individuals in many other natural populations, do not choose mates at random, population genetic theory and methods to account for assortative mating in empirical data remain largely underdeveloped. In the context of ancestry-assortative mating, extension of existing theory on how assortative mating shapes the expected distribution of a trait in a population (Wright 1921; Norris *et al.* 2019; Kim *et al.* 2021; Border *et al.* 2022; Muralidhar *et al.* 2022; Horwitz *et al.* 2023) is made difficult by ambiguity about what the relevant phenotype is. Furthermore, global ancestry proportion is an unusual quantitative phenotype, in that its trait value is integrated across every locus in the genome. How this genome-wide involvement might impact analyses, such as selection scans and association studies, that attempt to distinguish implicated loci from neutral loci remains unclear. Simulations of ancestry-assortative mating that accurately recapitulate key summaries of empirical data are crucial to unraveling these impacts. Here, we considered

2 related prerequisite questions: first, are there multiple mechanisms of mate choice compatible with the observed correlation in global ancestry proportion between spouses in human populations? Second, does the choice of a particular mathematical function for defining biased mating meaningfully impact the conclusions drawn from the resulting simulations? We compared 4 models, including one that considers social groups rather than quantitative similarity in global ancestry proportion as the mechanism of mate choice, to better understand how we ought to think about modeling and correcting for ancestry-assortative mating going forward.

We turn first to assumptions commonly made by statistical genetics methods (Zaitlen *et al.* 2017; Pfennig and Lachance 2023; Huang *et al.* 2024), which focus on the observed empirical correlation in global ancestry between mates: namely, that this correlation is constant over time and that parameters of interest are comparable between empirical data and simulated data with the same correlation coefficient. In our simulations, we find that $r(x_i, x_j)$ is not stable over time and decays toward zero under most models (Fig. 2). Thus, assuming that bias in mating is constant over time, the correlation in global ancestry proportion between spouses observed in a contemporary sample is likely

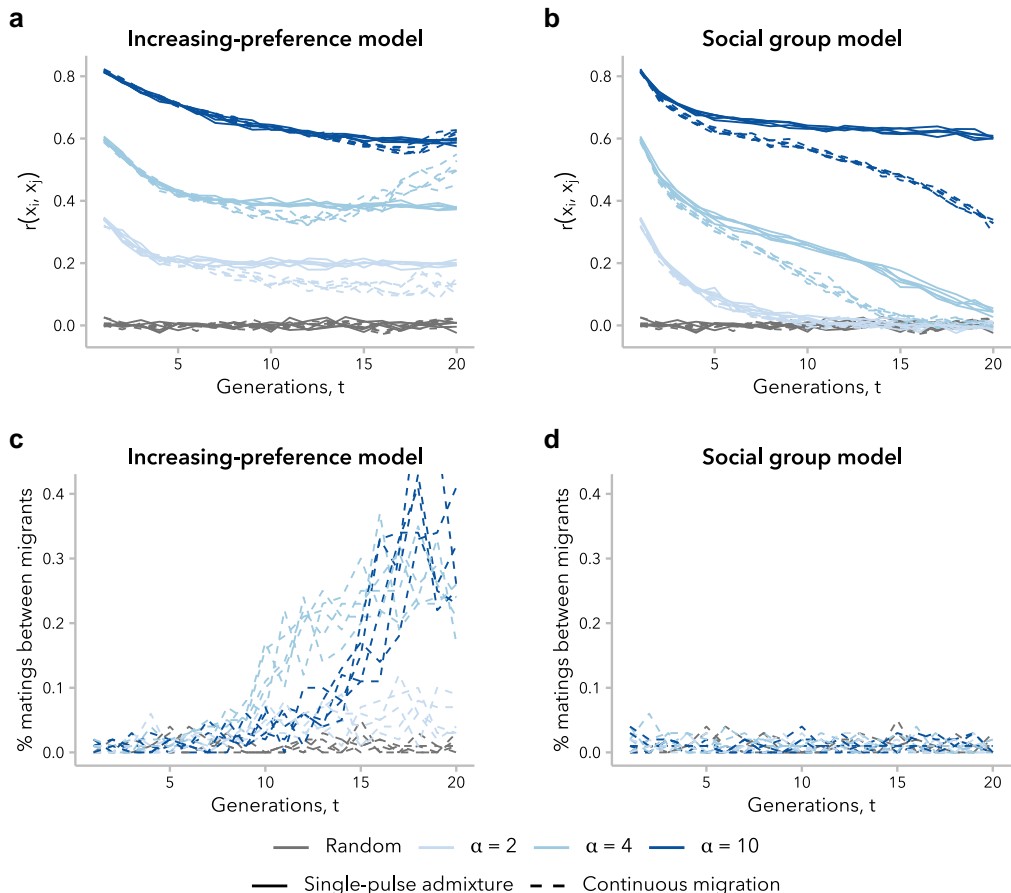

**Fig. 7.** The effects of continuous migration differed between mate choice models, driven by the percent of mating events between new migrants. a and b) Correlation in global ancestry proportion between mates, $r(x_i, x_j)$, increased over time in simulations under the increasing preference model with continuous migration, for some $\alpha$. This behavior was not observed in any simulations without migration (see Fig. 2) or under the social group model with continuous migration. c) Under the increasing preference model with continuous migration, mating between migrants was increasingly prevalent over time and far more frequent than expected under random mating. d) Mating between migrants under the social group model with continuous migration did not occur more often than expected by chance.

smaller than it was in previous generations. In addition, we find that multiple models can produce the same $r(x_i, x_j)$ while differing in the underlying mating structure (Fig. 4) and the distribution of local ancestry tract lengths (Fig. 5). However, we find that the effects of these differences between models on estimated time since admixture are likely to be small, likely because the effects on mean local ancestry tract length are similar (Fig. 6).

Prior studies that have modeled biased mate choice to develop theory about assortative mating have either not considered how the decrease in variance in global ancestry proportion across individuals over time impacts the correlation in global ancestry proportion between mates (Kim *et al.* 2021; Muralidhar *et al.* 2022) or explicitly modeled a constant correlation coefficient in the face of decreasing variance (Huang *et al.* 2024). Our work highlights that variance in global ancestry proportion across individuals plays an essential role in determining whether a positive correlation can be observed and, furthermore, whether mating is effectively random (Fig. 3; Supplementary Fig. 12). Prior theory has also prioritized the role of individual mate choice preference for similarity in global ancestry proportion ($|x_i - x_j|$). In humans particularly, there is a wide array of social science research to support the role of sociological factors in mating outcomes. To wit, we found that a simplistic model imposing a barrier to mating between 2 social groups was sufficient to generate signatures of

ancestry-assortative mating that could be observed for 20–50 generations post-admixture. To our knowledge, this type of model has not been used before to model ancestry-assortative mating in admixed populations but may be representative of how social categories like race, ethnicity, and socioeconomic status influence spouse choice.

The 4 models that we consider in the present study are not the only possible models but rather represent 2 classes of model worthy of further theoretical exploration: ancestry-similarity and social group models. For ancestry-similarity models, we have demonstrated that not all variants of this model type can sustain sufficiently high levels of variance in global ancestry proportion across individuals to continue to observe a positive correlation in global ancestry proportion between mates (at least, without additional in-migration events). The increasing preference model presents a mathematically simple strategy to compensate for this decay in $\sigma_x^2(t)$: increasing the choosiness of individuals. While this is effective in maintaining a high $r(x_i, x_j)$, it is likely unrealistic in practice, making individuals too attuned to small differences between mates (Supplementary Fig. 2b). This overcompensation is heightened in the continuous migration scenario, leading to unexpected (and undesirable) behavior (Fig. 7). Future development of ancestry-similarity models should consider alternative approaches to tune the selectiveness of

individuals over time. For instance, we might want to model a constant degree of bias (i.e. $\frac{\max(\psi_{i,j})}{\min(\psi_{i,j})}$) over time.

In including the social group model in this study, we aimed only to demonstrate that categorical social barriers can mediate ancestry-assortative mating, even when individuals do not directly use ancestry information to make mating decisions. To that end, we designed a proof-of-concept version of this model class, which is likely overly simplistic for drawing conclusions about real-world human populations. Future development of this class could include greater social complexity, for instance, more than 2 social groups, alternative rules for how individuals "inherit" their social group membership, and asymmetric barriers between groups. Each of these added layers could be implemented in different ways. For example, in a simulation with more than 2 groups, barriers might be more permeable between some pairs of groups than others.

A growing body of research suggests that patterns of assortative mating are not stable over time (e.g. Mare 1991; Sunde *et al.* 2024). Under an ancestry-similarity model, this could reflect changes in preference over time, potentially to account for decreasing differences between potential mates (as discussed above). Under a social group model, this could reflect changes in social mobility or acceptance of intergroup mating. Thus, modeling this type of change over time would be an interesting future direction for both classes of model, although it remains outside the scope of the present work.

Here, we focus exclusively on biased mate choice as a mechanism for generating a positive correlation in global ancestry proportion between mates, comparing potential models for implementing that mechanism. Future work should also consider other mechanisms that require non-Wright–Fisher frameworks to incorporate additional parameters (e.g. birth and death rates, dispersal rates, etc.). For instance, in real-world populations, geographic structure (i.e. isolation by distance over continuous space) is also a major driver of nonrandom mating with respect to ancestry. It remains an open question whether geography alone could produce patterns of assortative mating that resemble empirical observations, as well as how geography might interact with social barriers to shape the opportunities for individuals to encounter one another and act on biased mate preferences.

Taken together, our results emphasize that theory incorporating nonrandom mating must carefully consider how mate choice is conceptualized and modeled. While our focus is primarily on how to better model ancestry-assortative mating, we also make a few recommendations for future empirical studies of ancestry-assortative mating in humans. First, observing a correlation in global ancestry proportion between spouses can reflect multiple mechanisms of mate choice and should not be interpreted as unequivocal evidence of a preference for mates with similar global ancestry proportion. Second, the magnitude of the correlation coefficient is limited by variance in global ancestry proportion across individuals. Thus, a positive correlation in global ancestry proportion in a real-world population should be interpreted as specific to the time point sampled. Third, there are multiple mating structures that can give rise to the same Pearson correlation coefficient, and visualization of these patterns (e.g. with a hexagonal bin plot or hurricane plot of $\Delta_x$) may help to disambiguate these possibilities.

As with any simulation study, our results are limited by what we have elected to model or not model. We highlight 2 important caveats. First, we do not model genetic variation within and between source populations. Empirical analyses rely on these data to first infer global and local ancestry, a process that is sensitive to how researchers define discrete source populations from continuous human genetic variation. Having bypassed this step, we cannot comment on how estimation error might interact with our results. Second, we have focused on a single pulse of admixture, turning to a continuous migration scenario only to highlight an unexpected result from the increasing preference model. More realistic models of human populations almost certainly involve complex migration dynamics, including multiple pulses of migration, differing contributions from source populations, changes in population size, asymmetries in mate preferences and in the ability of individuals to enact their mate choices, monogamous mating, and multiway admixture. Inclusion of these additional factors into future models is likely to impact the results. However, each additional factor drastically expands the range of possible implementations and space of parameter values. As such, these more complex models are likely to be most useful when targeted to matching the parameters of a population of interest whose history is well understood and less well suited to a general exploration of parameter space.

## Data availability

All scripts used for simulations, analyses, and figures are available on GitHub at: https://github.com/agoldberglab/ancestry-assortative-mating-simulation.

Supplemental material available at GENETICS online.

## Acknowledgments

We thank Joshua G. Schraiber (ORCiD: 0000-0002-7912-2195), Michael M. Hoffman (ORCiD: 0000-0002-4517-1562), John Barton (ORCiD: 0000-0003-1467-421X), Shyamalika Gopalan (ORCiD: 0000-0002-2608-8472), and members of the Goldberg lab for helpful discussions.

## Funding

This work was supported by the National Institute of General Medical Sciences (R35 GM133481 to AG and R35 GM146926 to ZAS).

## Conflicts of interest

The author(s) declare no conflicts of interest.

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

*Editor: S. Gravel*