## [Peer Review File · Genetics]

Differentiating mechanism from outcome for ancestry-assortative mating in admixed human populations

Dashiell Massey, Zachary Szpiech, and Amy Goldberg

NOTE: The reviews and decision letters are unedited and appear as submitted by the reviewers.

In extremely rare instances and as determined by a Senior Editor or the EIC, portions of a review may be redacted. If a review is signed, the reviewer has agreed to no longer remain anonymous.

The review history appears in chronological order.

Review Timeline:

Submission Date:	2024-06-06
Editorial Decision:	2024-08-21
Resubmission Received:	2024-12-19
Accepted:	2025-01-30

August 20, 2024

GENETICS-2024-307165

Differentiating mechanism from outcome for ancestry-assortative mating in admixed human populations

Dear Dr. Goldberg:

Two experts in the field have reviewed your manuscript, and I have read it as well. We all appreciated the relevance of the topic and the interest and quality of the analyses presented. While your manuscript is not currently acceptable for publication in GENETICS, we would welcome a substantially revised manuscript. Both reviewers have comments and concerns to be addressed in a revised manuscript. You can read their reviews at the end of this email.

I think it would be most helpful to make the work a bit more connected to empirical work, ideally by analyzing real data (or discussing challenges to doing so), as suggested by reviewer 1, or by showing how your simulated populations would look like using typical measures used in empirical studies. We look forward to receiving your revised manuscript. Please let the editorial office know approximately how long you expect to need for revisions.

Upon resubmission, please include:

1. A clean version of your manuscript;
2. A marked version of your manuscript in which you highlight significant revisions carried out in response to the major points raised by the editor/reviewers (track changes is acceptable if preferred);
3. A detailed response to the editor's/reviewers' feedback and to the concerns listed above. Please reference line numbers in this response to aid the editor and reviewers.

Your paper will likely be sent back out for review.

Additionally, please ensure that your resubmission is formatted for GENETICS
<https://academic.oup.com/genetics/pages/general-instructions>

Follow this link to submit the revised manuscript: Link Not Available

Sincerely,

Simon Gravel
Associate Editor
GENETICS

Approved by:
Nicholas Barton
Senior Editor
GENETICS

Additional comments by Associate editor (SG):

I appreciated the careful analysis of an important problem, and also read the paper with interest. In addition to the points raised by the reviewers, I would like to highlight two points worth considering.

1)
I would have been interested in seeing more discussion of the role of geographical structure. Geographical structure is a typical baseline null model when considering whether ancestry-assortative mating has taken place (see e.g. the introduction in Sebro et al, 2017). The social structure model simulated here seems to be almost indistinguishable from geographical structure, and it seems like this would be an important conceptual point to address, for two reasons.

First, social substructure and geographical isolation can be quite different demographic processes (They can also be closely intertwined, as in geographic segregation). Since I feel like the purpose of this paper is to carefully pick apart possible causal mechanisms for ancestry correlations, I think this deserves some discussion.

For example, it would be helpful to discuss whether you intend to conceptually distinguish social and geographic isolation, or whether you consider them as two sides of the same coin.

If you see them as the same, I think it would be important to clarify the relationship of your model and interpretation to that of

previous work addressing geographical structure. E.g. at l.621, to claim novelty I think you would need to be able to distinguish social and geographic barriers.

Empirically, a geographically structured model can sometimes be observed directly on a PCA plot (e.g. Sebro et al 2017 again), or on an admixture plot. If it is possible to show some of these in your simulated models, it may be also help empiricists understand just how structured your simulations are.

2)

I am not clear from reading the paper whether there is any social science evidence backing the increased preference model, or whether this is purely a mathematical exercise of seeing what, in theory, could explain the data. If the latter, I would consider wrapping it in a thicker layer of caveats, since you are speculating on race-correlated social behaviour for a mathematical exercise.

I do agree with highlighting that preferences can change over time. Maybe you can simply present this model as a particularly extreme and mathematically tractable example of changing preferences used for illustration, rather than as a plausible alternate model (if you agree with this interpretation).

l. 235, 273: alpha seems defined twice differently
l.381: extra word

Reviewer #1 (Comments for the Authors (Required)):

In this paper, the authors use simulations to explore several classes of assortative mating models in order to understand how the mechanism of mate choice shapes outcomes which can be measured from data, such as the correlation in ancestry between mates and the distribution of ancestry proportions for a given number of generations since the initialization of admixture. The authors emphasize that current approaches conflate admixture mechanism and outcome, leading to misestimates of admixture age and an overemphasis on correlation coefficients as a summary of historical admixture dynamics. Their results demonstrate that under the standard approach of analyzing ancestry correlations, distinct assortative mating mechanisms are indistinguishable.

In general, this is a rigorous manuscript, its conclusions are well founded, and it is a valuable contribution to theoretical population genetics. The introduction in particular is well referenced and compelling. However, there is room to improve the clarity of its methods and the communication of the impact of its results. The models section provides a nice overview of several existing models for mate preference. However, the model parameters are ill defined and Figure 1 seems to contain erroneous plots of the model equations. The results section is thorough, but at times loses focus. Part of this manuscript's novelty comes from its synthesis of many models, which is an important contribution to the field. However, this framing necessitated many model comparisons and a profusion of corresponding main-text and supplemental figures, which sometimes left the reader digging for the most important results in a sea of analyses. Additionally, the proposed statistic, expressed mating bias B , would benefit from more motivation and a demonstration with real data if it is intended as an alternative to existing variance or correlation based methods to understand assortative mating.

Major comments:

1. Please clarify the main goals of the paper. It is clear that you are highlighting ambiguities present in current analysis measures, but are you introducing an alternative? If the goal is to introduce the expressed mating bias B as an alternative, more time should be devoted to motivating and demonstrating B , potentially through an example data analysis.
2. Relatedly, please include a stronger focus throughout the results section on how the results connect to data analysis. I appreciated the guidance in the discussion section (lines 637-647), but the results section could be strengthened by including recurring references to data applications, or even an example data analysis. If a data analysis example is included, please replicate the key figures from the results section with real instead of simulated data, and advise the reader on their interpretation. This addition would enable the reader to compare data to simulation results and to understand the challenges of inferring the assortative mating histories of real populations.
3. It was sometimes difficult to identify the most important results of the paper. Key result statements felt buried in the middle of paragraphs. Please consider streamlining the results section and moving some content to a supplemental section so that the main text results section is focused on the most important results.
4. Language used to refer to key concepts across the manuscript was sometimes ill-defined and inconsistent. Please ensure that

the following key concepts have clear mathematical and conceptual definitions early in the paper, and please use consistent language and notation when referring to the same concept:

- a. Global ancestry, global ancestry proportion, ancestry proportion, x_i , x (p. 11)
 - b. Correlation in global ancestry proportion between spouses/mating pairs, correlation in ancestry, $r_{t=1}$ (p. 8), r (used in panel labels of Figure 3 and many supplemental figures)
 - c. "Generation-specific population variance in ancestry, σ_g^2 " (p. 5), (population) variance in ancestry proportion (p. 12-13)
5. Relatedly, mathematical notation of model parameters was also inconsistent, especially the mate-choice strength parameters used in the model equations and most main-text figures. Please make this notation more clear and consistent by addressing the following points.
- a. Please number important equations and reference these equation numbers when mathematical properties of the models are discussed, for example in the caption of Figure 1. As written, it is sometimes ambiguous which equation is being referenced.
 - b. Please consistently define all model parameters when they are introduced.
 - i. When introducing models and their parameters (e.g., x_i , s_i , ψ_i , c , α , σ), please include their permissible range (e.g., " $c > 0$ " or " $x_i \in [0, 1]$ ") and any other important properties.
 - ii. For mate-choice preference parameters (c , σ , α), specify the limit of each parameter in which mate choice is random (i.e., does mate choice get more or less random as the parameter gets larger?). For example, in Figure 1, please specify the value (or limit) of α corresponding to random mating.
 - c. Please correct the figures which use α as the mate-choice parameter for all models, including those parameterized with c or σ . If a reparameterization of these models in terms of α is used in place of the equations presented in the models section, please make this clear. Specific comments for Figure 1 are included in the Models section below.

Minor comments:

1. Please consistently specify the objects correlations or variances are computed across, i.e., "variance of X" or "correlation between X and Y."
 - a. There were also instances where the object was specified, but ambiguity remained. For example, in the caption of Figure 6 and throughout the corresponding section (lines 455-502), it was unclear if ancestry correlations are of ancestry proportions or ancestry tract lengths.
2. At times, sections felt disjointed and lacked an overarching narrative. Interesting mathematical objects, such as the discussion of A versus α (lines 267-283) and the expressed mating bias B (line 430), were often introduced but not discussed in other sections, isolating them from the broader analysis and undermining the motivation for their inclusion in the paper.
3. The study relied on simulations, but would be strengthened by analytical results. For example, I expect one could derive the analytic relationship between the variance of ancestry proportions or the correlation between mates' ancestry proportions, and the expressed mating bias B. These relationships could define the curves along which points fall in Figure 5, providing both a more precise understanding of these variables' relationships and an independent confirmation of the simulation results. Or, if the permutation aspect of B's definition makes such an analytical result intractable, it would be interesting to explore the relationship between correlation and variance in order to understand the driver of the greater difference between models in Figure 5B than in 5A.

Introduction:

1. Please include a brief discussion of within- versus across-group variability, e.g., Lewontin (1972) and related work. Please be mindful of the potential for statements about genetic differences among human populations to be exaggerated or taken out of context.

Models:

1. Figure 1: The model equations do not match their corresponding plots in Figure 1.
 - a. The stationary-preference model equation (line 169) and the increasing-preference model equation (line 182) are both functions of c and x_i , and have an upper bound of 1. The corresponding plots (panels A and B) are a function of α and x_i , and have an upper bound of 2.
 - i. Please correct whichever is mistaken, or explain if panels A-B are not plotting the equations in line 169 and 182, respectively.
 - ii. Please also clarify if panel B is using the infinite-sites model introduced in line 184.
 - b. The broad-preference model (line 197) has neither α or c , but instead is parameterized by x_i and "a fixed parameter σ " which is never defined (line 196). The corresponding plot (panel C) again is a function of x_i and α .
 - i. Please specify the values of σ used for panel C, and correct the figure legend.
 - c. The social group model does have an α parameter, but the values plotted in panel D do not match the values obtained when you plug $\alpha=4$ or 10 into the equation at line 235.
 - i. Please correct the values in panel D or the equation at line 235.
 - d. Please explain what is meant by "weighted sampling" (line 258).
2. Please number important equations and reference those numbers in addition to the model names.
3. Please consistently define all model variables when they are introduced. See Major Comments 4 and 5.
4. It seems that much previous work relied on a framework with two distinct sexes, with one being more selective. Please

mention (either here or in the discussion) if you expect your simulation results to depend on this choice.

5. Please discuss (either here or in the discussion) how the results are shaped by the genetic diversity within populations and the similarity across source populations.
6. Lines 136-141: Please explain how x_i and s_i are derived from the SLiM simulated genome.
 - a. Please also clarify if the models assume the presence of only two ancestry groups, with proportion $(x_i, 1-x_i)$ respectively in individual i .
7. Lines 170-178: Why do you choose a focal individual x_1 and compute ψ_i , rather than computing a more general, pairwise weight $\psi_{i,j}$?
8. Lines 179-190: Please define the generation-specific population variance in ancestry σ_g^x in terms of both g and x . Is the x here the mean ancestry proportion across the population at generation g ?
9. Line 196: Please define the "fixed parameter, σ ." If σ is not related to the variance in ancestry, please use a different symbol for this parameter.
10. Lines 207-210: Please clarify whether the categorical trait in these examples is species identity.
11. Lines 231-235: As discussed in Major Comment 5, please include the permissible range of α and ψ_i . Please also include the cases when ψ_i is minimized or maximized, and the interpretation of these cases (e.g., "when $\alpha=0$, ψ_i takes its maximal value of 1, implying that...").
12. Lines 267-284: The purpose of this subsection is unclear. How does this section and its α parameter relate to the previous models and their c , σ , and α parameters? Why is the $r_{t=1}$ notation used here but nowhere else, and is this the same correlation discussed elsewhere? If this subsection's purpose is to provide an empirical definition of the mate choice strength parameter used in the earlier models, then this section should be included at the beginning of the models section rather than at the end.
 - a. Line 268: please include parenthetical references to the mate choice function equation numbers.
 - b. Lines 275-280: I appreciate this explanation of the mating bias parameter, α , but it needs to be far earlier in the models section (e.g., lines 231-235, or earlier if it applies to the c parameter used in the continuous models as well).
 - c. Lines 280-284: This is quite an abrupt introduction to $r_{t=1}$.
 - i. Please ease the reader into this explanation by inserting a preceding sentence that defines and motivates the correlation in mate ancestry.
 - ii. Please provide more intuition for the equation at line 284. For example, at the moment it is unclear where the 0.25 is coming from.
 - iii. Why do you introduce the notation for " $r_{t=1}$ " here but not use it anywhere else?

Results:

1. Lines 327 - 368: Please describe the simulations' initial ancestry proportion distribution.
2. Figure 3: Would it be helpful to plot the line $y=x$ on these plots as a guide to the eye? Also, I assume these plots should be symmetric about $y=x$; are asymmetries just due to sampling noise?
3. Lines 415-433: Please define $\bar{\Delta}_x$ and $\bar{\Delta}_{\text{permuted}}$. As discussed above, please also consider expanding this section if B is intended to be used as a summary statistic to understand assortative mating in real data.

Reviewer #2 (Comments for the Authors (Required)):

This paper is a simulation study of various models of ancestry-based mate choice in admixed populations. The models are in two classes: "ancestry similarity" models where the probability of mating depends directly on the similarity of individuals' ancestries, and a "social group" model where individuals are more likely to mate if they are in the same social group (with distinct ancestries initially occupying separate social groups).

The strength of assortative mating is usually measured by the correlation coefficient between mates. The authors find that commonly used ancestry-similarity models of assortative mating cannot maintain a fixed positive ancestry correlation between mates in the generations after admixture. The reason is that, after admixture, recombination rapidly mixes together the two ancestries in the population, reducing variance among individuals in their ancestry fractions. As individuals become more ancestry-similar, they become less discriminatory in mate choice under these standard models. As the authors show, the situation can be rescued by an "increasing-preference" ancestry-similarity model in which the strength of the preference increases as the average ancestry differences among individuals decline---i.e., individuals become more discriminatory in mate choice.

The authors also show that a given ancestry correlation coefficient among mates can be achieved via both ancestry-similarity and social-group models; this finding complicates mechanistic interpretations of such correlation coefficients.

In general, I found the results in the paper interesting, and they certainly add some nuance to how we commonly think of measuring and understanding assortative mating. I have one major comment on the authors' key findings, and a few more minor comments, all detailed below.

MAJOR COMMENT

The authors' primary goal is to aid our understanding of the etiology and consequences of ancestry-based assortative mating in humans. However, the mating structure that they implement using the "mateChoice" callback in their SLiM simulations allows individuals to mate an unlimited number of times within a generation (see lines 142-147). In reality, the variance across humans in their reproductive success is more limited.

This difference between the authors' model and the mating structure of human populations could potentially cause large differences in ancestry dynamics under various kinds of assortative mating. The reason is that, with polygamous mating, models of assortative mating can in some cases induce strong sexual selection in favor of one ancestry or the other. In the authors' simulations, neither ancestry can be systematically favored by sexual selection because they start as equal fractions of the population, but the possibility of sexual selection then makes this (unrealistic) assumption of precisely equal starting fractions crucial.

So I think it is important for the authors to show that their results are not quantitatively driven by the potentially large variances in reproductive success among individuals that can arise in their model. The simplest way to do this would be to enforce monogamy in the simulations, with each individual mating only once, and each mating pair producing two offspring. This scenario is possible, if probably a little tedious, to simulate using the non-Wright-Fisher environment in SLiM (see e.g. Veller & Coop 2024). I confess that I am not entirely sure how to port the authors' functional forms for mate choice to this monogamous scenario, in which mates must be picked without replacement rather than with replacement, but given that monogamy is a more realistic scenario for many human populations than the authors' scenario of unlimited polygamy, I think it is an important exercise to carry out.

(Alternatively, the authors could (i) directly measure the variance in reproductive success among individuals in their simulations, and check if these are very dissimilar to values measured in humans, and (ii) carry out some simulations with unequal starting fractions of the two ancestries, to see if sexual selection is induced in such cases.)

MINOR COMMENTS

lines 151-153: "... (Lande 1981; Kirkpatrick 1982; Seger 1985). "Generally, under these models one locus controls the male phenotype while a second locus controls the female preference for that phenotype."
This is not true for quantitative genetic models of sexual selection such as the cited Lande 1981 paper and the large literature it inspired.

line 181: Why sigma for the variance, and not sigma-squared as usual?

lines 415-433: To complement the usual metric for the strength of assortative mating, the correlation coefficient among mates, the authors suggest a new statistic which they call B. B is based on the average absolute distance in ancestries among mates $|x-y|$; specifically, it is $1 - [\text{the ratio of the average value of } |x-y| \text{ under assortative mating to its average value if mate choice were random, as computed by the authors using a bootstrap}]$ (line 430). However, this statistic B is mathematically almost the same thing as the correlation coefficient, as can be seen if we compute the same metric but replace the absolute value $|x-y|$ with the square $(x-y)^2$.

Call this analogous metric B'. Under assortative mating, $E[(x-y)^2] = \text{Var}(x) + \text{Var}(y) - 2\text{Cov}(x,y) = 2[\text{Var}(x) - \text{Cov}(x,y)]$ (since x and y are identically distributed). Under random mating, $\text{Cov}(x,y) = 0$, so $E[(x-y)^2] = \text{Var}(x) + \text{Var}(y) = 2\text{Var}(x)$. So $B' = 1 - [\text{Var}(x) - \text{Cov}(x,y)]/\text{Var}(x) = \text{Cov}(x,y)/\text{Var}(x) = \text{Cov}(x,y)/\sqrt{[\text{Var}(x)\text{Var}(y)]}$, the correlation coefficient $\rho(x,y)$.

This close mathematical correspondence between B and the correlation coefficient presumably explains why they behave almost identically in the authors' simulations (e.g., Fig. 5A).

Fig. 7: I found this figure unnecessarily difficult to understand. There are no x-axis labels in panels A and B, and I can't actually tell what these should be. There are two sets of dashed lines in each panel A and B, but only one is referred to in the caption (in general, these panels would benefit from direct labelling rather than just descriptions in the caption). Also, could A and B not be in a single panel with the same axes? Panels D and E should not be described as such in the caption; instead, their description is contained in that of panel C.

line 571-573: "global ancestry proportion ... is determined by every locus in the genome, meaning that the entire null distribution of any parameter of interest is likely to be affected [by assortative mating]."

I didn't understand this.

lines 595-597: This seems unfair to the cited papers (one of which is co-authored by the senior author of the present manuscript!). Those papers are, in fact, correct to "assume that [biased-mate choice] models are sufficient to generate a correlation in ancestry", and, though they would have been incorrect to assume that the models are sufficient to generate a constant or persistent correlation in ancestry, they did not, as far as I can tell, make such an assumption.

TYPOS/WORDING

line 63: "have suggested temporal structure"  "have suggested that temporal structure"

line 435: "that correlation coefficient"  "that the correlation coefficient"

line 455: "produced in an excess"  "produces an excess"

line 630: "patterns .. is"  "patterns ... are"

Associate Editor Comments:

We thank the reviewers and editor for their thoughtful comments. Major revisions in this resubmission include:

(1) **Restructuring of the Models section** to first highlight the underlying approach shared across the models and then provide more detail on each model individually.

(2) **Changes to the order of subsections in the Results section** to more clearly delineate when we shift our focus from a general comparison of the four models to a targeted analysis of paired simulations under the increasing-preference vs. social group models.

(3) **Addition of Supplemental Figures S14 and S15** to alleviate concerns about the interaction of biased mate-choice with reproductive success.

Further specific edits are described below, with larger changes shown in tracking or comments in the text.

For clarity, we have numbered the comments from the editor & reviewers sequentially. Line numbers reference the “clean version” of the revised text (without tracked changes). We note in individual responses when figure numbers have changed relative the previous version.

Additional comments by Associate Editor (SG):

I appreciated the careful analysis of an important problem, and also read the paper with interest. In addition to the points raised by the reviewers, I would like to highlight two points worth considering.

1. I would have been interested in seeing more discussion of the role of geographical structure. Geographical structure is a typical baseline null model when considering whether ancestry-assortative mating has taken place (see, *e.g.*, the introduction in Sebro *et al.*, 2017). The social structure model simulated here seems to be almost indistinguishable from geographical structure, and it seems like this would be an important conceptual point to address, for two reasons.

First, social substructure and geographical isolation can be quite different demographic processes. (They can also be closely intertwined, as in geographic segregation.) Since I feel like the purpose of this paper is to carefully pick apart possible causal mechanisms for ancestry correlations, I think this deserves some discussion.

For example, it would be helpful to discuss whether you intend to conceptually distinguish social and geographic isolation, or whether you consider them as two sides of the same coin. If you see them as the same, I think it would be important to clarify the relationship of your model and interpretation to that of previous work addressing geographical structure. *E.g.*, at **line 621**, to claim novelty I think you would need to be able to distinguish social and geographic barriers.

Empirically, a geographically structured model can sometimes be observed directly on a PCA

plot (e.g., Sebro *et al.*, 2017 again), or on an admixture plot. If it is possible to show some of these in your simulated models, it may also help empiricists understand just how structured your simulations are.

We completely agree that limited dispersal across geography can also produce ancestry-associated mating patterns. This is a separate process from the types of mate-choice bias we model here, and it seems likely that both processes act in concert to shape empirical observations for human spouse pairs. We have added a paragraph to the Discussion (**lines 643-652**) describing what we see as promising future directions beyond mate-choice bias, focusing on geography alone and in combination with biased mate preference. Modeling geographic isolation requires more complex simulations than we present here, using a non-Wright-Fisher framework to consider how to model the dispersal of individuals from their place of birth and the geographic radius within which they seek a mate.

We have also added a sentence in the Models section to state explicitly that we did not model geographic space (**lines 140-141**).

The novelty of our social group model is its demonstration that a model of *categorical* mate-choice bias can produce similar results to a model in which mate-choice is biased by a continuous function. (Isolation-by-distance would also be continuous.) To clarify this point, we have added the word “categorical” in **line 627** to that effect.

Finally, in the present work, we only model local-ancestry tracts, from which we can calculate global ancestry proportion (clarified in **lines 142-145**). We do not model any mutational processes or genetic variation within or between source populations, which would require assumptions about the evolutionary history of the source populations and admixture process. Thus, we do not have the type of simulated data required for the suggested additional visualizations (PCA and admixture plot).

2. I am not clear from reading the paper whether there is any social science evidence backing the increased preference model, or whether this is purely a mathematical exercise of seeing what, in theory, could explain the data. If the latter, I would consider wrapping it in a thicker layer of caveats, since you are speculating on race-correlated social behavior for a mathematical exercise.

I do agree with highlighting that preferences can change over time. Maybe you can simply present this model as a particularly extreme and mathematically tractable example of changing preferences used for illustration, rather than as a plausible alternate model (if you agree with this interpretation).

There is some evidence (cited in **line 637**) that mating preferences (or at least, mating behaviors) change over time. However, the increasing-preference model was originally a mathematical approach by Kim *et al.* to maintain certain mathematical properties of the ancestry distribution rather than model the underlying process realistically (we have clarified this point in **lines 212-215**). More generally, our goal is to examine the behavior of models that *are already being used in the literature*, which may or may not be reasonable approximations of realistic mating scenarios.

We agree that strong caution in interpretation about human behavior is warranted; instead, our goal is to highlight the myriad ways people are currently modeling ancestry-assortative mating and demonstrate that differences between these models have important consequences for theory on mate choice. In the discussion, we highlight that the increasing-preference model is “mathematically simple” (**line 620**) and “likely unrealistic” (**line 621**), and that it overcompensates for this decreasing variance in global ancestry proportion (**lines 623**).

MINOR COMMENTS:

3. **lines 235 + 273**: α seems defined twice differently

After some rearrangement of the Models section, the two definitions of α referenced in this comment are found in EQUATION 1 (**line 171**) and EQUATION 5 (**line 264**). We have added a sentence in **line 264** to say that EQUATION 5 is a rearrangement of EQUATION 1.

4. **line 381**: extra word

This sentence (and much of surrounding paragraph, starting **line 350**) has been rewritten in the revised text for greater clarity.

Reviewer #1:

In this paper, the authors use simulations to explore several classes of assortative mating models in order to understand how the mechanism of mate choice shapes outcomes which can be measured from data, such as the correlation in ancestry between mates and the distribution of ancestry proportions for a given number of generations since the initialization of admixture. The authors emphasize that current approaches conflate admixture mechanism and outcome, leading to misestimates of admixture age and an overemphasis on correlation coefficients as a summary of historical admixture dynamics. Their results demonstrate that under the standard approach of analyzing ancestry correlations, distinct assortative mating mechanisms are indistinguishable.

In general, this is a rigorous manuscript, its conclusions are well founded, and it is a valuable contribution to theoretical population genetics. The introduction, in particular, is well referenced and compelling. However, there is room to improve the clarity of its methods and the communication of the impact of its results. The models section provides a nice overview of several existing models for mate preference. However, the model parameters are ill-defined, and **Figure 1** seems to contain erroneous plots of the model equations. The results section is thorough, but at times loses focus. Part of this manuscript's novelty comes from its synthesis of many models, which is an important contribution to the field. However, this framing necessitated many model comparisons and a profusion of corresponding main-text and supplemental figures, which sometimes left the reader digging for the most important results in a sea of analyses. Additionally, the proposed statistic, expressed mating bias B , would benefit from more motivation and a demonstration with real data if it is intended as an alternative to existing variance or correlation-based methods to understand assortative mating.

We appreciate the positive comments on the introduction and results of the manuscript. We discuss below several large changes to clarify our goals, methods, and results sections.

MAJOR COMMENTS:

5. Please clarify the main goals of the paper. It is clear that you are highlighting ambiguities present in current analysis measures, but are you introducing an alternative? If the goal is to introduce the expressed mating bias B as an alternative, more time should be devoted to motivating and demonstrating B , potentially through an example data analysis.

We clarify a main conclusion in the discussion (**lines 653-664**). The main goal of this paper is to highlight that we still do not have a good understanding of why positive correlations in global ancestry proportion between spouses have been repeatedly observed in data from human populations. Namely, we demonstrate that multiple generative mechanisms can explain empirical observations equally well. For the empiricist reader, the take-home message is that these correlations have multiple plausible interpretations and should not be viewed as strong support for a particular view of mating practices. For the theoretician, we emphasize that the choice of mating model and its impact on downstream analysis deserves careful consideration.

We have removed the discussion of B , which as Reviewer 2 points out is closely related to the correlation coefficient (see Comment #31 below).

6. Relatedly, please include a stronger focus throughout the results section on how the results connect to data analysis. I appreciated the guidance in the discussion section (**lines 637-647**), but the results section could be strengthened by including recurring references to data applications, or even an example data analysis. If a data analysis example is included, please replicate the key figures from the results section with real instead of simulated data and advise the reader on their interpretation. This addition would enable the reader to compare data to simulation results and to understand the challenges of inferring the assortative mating histories of real populations.

As outlined above, our primary objective in this manuscript is to communicate that there is a gap in our current understanding of ancestry-assortative mating in humans. We do this with two audiences in mind, seeking to highlight an underappreciated interpretational ambiguity for readers who work with empirical data, while also demonstrating how existing models fall short for theory-driven readers. We describe in the Discussion how our results provide us with ideas about how future models might innovate on existing theory in productive directions (**lines 643-652**).

We aim to caution readers against overinterpreting empirical data, especially the commonly seen correlation plots between ancestry of mating pairs. This makes an empirical example not illustrative. Instead, because the key conclusion regarding empirical data analysis is that there are multiple generative processes that can lead to similar observations which are only sometimes distinguishable, we can only make this point with simulated data where we know the underlying truth.

7. It was sometimes difficult to identify the most important results of the paper. Key result statements felt buried in the middle of paragraphs. Please consider streamlining the results section and moving some content to a supplemental section so that the main text results section is focused on the most important results.

We thank the reviewer for their helpful feedback that improved the clarity of our paper. We have substantially restructured the results section to clarify the main takeaways. Specifically, we have changed the order of subsections within the results to mark a clearer separation between the first part (comparing all four models) and the second part (comparing matched pairs of simulations under the increasing-preference and social group models that resulted in the same $r(x_i, x_j)$ value). We have also revised the titles of these subsections to provide a more concrete statement of the main result for the associated figure and deleted the section on B .

8. Language used to refer to key concepts across the manuscript was sometimes ill-defined and inconsistent. Please ensure that the following key concepts have clear mathematical and conceptual definitions early in the paper, and please use consistent language and notation when referring to the same concept:

- a. Global ancestry; global ancestry proportion; ancestry proportion; x_i ; x (p. 11)

We have revised the text to consistently refer to global ancestry proportion, x , which is defined on **line 145**. We use x_i and x_j to refer to the global ancestry proportions of the individuals i and j , respectively.

- b. Correlation in global ancestry proportion between spouses/mating pairs; correlation in ancestry; $r_{t=1}$ (p. 8); r (used in panel labels of Figure 3 and many supplemental figures)

We have revised the text to consistently refer to the correlation in global ancestry proportion between mates, $r(x_i, x_j)$. This is first defined on **line 179**.

- c. Generation-specific population variance in ancestry, σ_g^x (p. 5); (population) variance in ancestry proportion (p. 12-13)

We have revised the text to consistently refer to the variance in global ancestry proportion across individuals at time t , $\sigma_x^2(t)$. This is first defined on **line 215**.

9. Relatedly, mathematical notation of model parameters was also inconsistent, especially the mate-choice strength parameters used in the model equations and most main-text figures. Please make this notation more clear and consistent by addressing the following points:
 - a. Please number important equations and reference these equation numbers when mathematical properties of the models are discussed, for example in the caption of **Figure 1**. As written, it is sometimes ambiguous which equation is being referenced.

We have numbered the equations and include those equation numbers in the caption for **Figure 1**.

- b. Please consistently define all model parameters when they are introduced:
 - i. When introducing models and their parameters (e.g., $x_i, s_1, \psi_i, c, \alpha, \sigma$), please include their permissible range (e.g., $c > 0$ or $x_i \in [0,1]$) and any other important properties.

We have included the permissible range for each parameter as it is introduced.

- ii. For mate-choice preference parameters (c, σ, α), specify the limit of each parameter in which mate choice is random (i.e., does mate choice get more or less random as the parameter gets larger?). For example, in **Figure 1**, please specify the value (or limit) of α corresponding to random mating.

We have rearranged the Models section so that the common mate-choice parameter α is defined earlier. We explain how to interpret the magnitude of α in **lines 173-176**. For c and σ^2 , we have added an explanation of how each parameter relates to α on **lines 201 and 226**, respectively.

- iii. Please correct the figures which use α as the mate-choice parameter for all models, including those parameterized with c or σ . If a reparameterization of these models in terms of α is used in place of the equations presented in the models section, please

make this clear. Specific comments for **Figure 1** are included in the Models section below.

We apologize for the confusion regarding **Figure 1**. We have expressed EQUATION 3 and EQUATION 4 in terms of c to make clear that they are exponential decay functions, and EQUATION 5 in terms of σ^2 to make clear that it is the normal distribution function. However, we re-parameterize each of the models in terms of α in order to compare between models.

To clarify this, we have moved up the explanation of α to **line 166** and explicitly stated that $c = \ln(\alpha)$ and $\sigma^2 = \frac{1}{\sqrt{-2\ln(\alpha^{-1})}}$.

MINOR COMMENTS:

10. Please consistently specify the objects correlations or variances are computed across, *i.e.*, “variance of X ” or “correlation between X and Y .” There were also instances where the object was specified, but ambiguity remained. For example, in the caption of **Figure 6** and throughout the corresponding section (**lines 455-502**), it was unclear if ancestry correlations are of ancestry proportions or ancestry tract lengths.

We have clarified this by using more consistent phrasing throughout the manuscript (see response to Comment #8). For example, in the referenced section (now **lines 490-509** and **Figure 5**), we are referring to the correlation in global ancestry proportion between mates, $r(x_i, x_j)$, and not the correlation in local-ancestry tract lengths.

- a. At times, sections felt disjointed and lacked an overarching narrative. Interesting mathematical objects, such as the discussion of A versus α (**lines 267-283**) and the expressed mating bias B (**line 430**), were often introduced but not discussed in other sections, isolating them from the broader analysis and undermining the motivation for their inclusion in the paper.

We have removed the discussion of both A and B . While interesting asides, the reviewer is correct that neither is relevant to the main narrative of the manuscript.

To address this comment more broadly, we have streamlined the methods and results sections in response to multiple comments, most relevantly, changing the order of subsections in the Results and improved the titles of the results subsections.

11. The study relied on simulations but would be strengthened by analytical results. For example, I expect one could derive the analytic relationship between the variance of ancestry proportions or the correlation between mates' ancestry proportions, and the expressed mating bias B . These relationships could define the curves along which points fall in **Figure 5**, providing both a more precise understanding of these variables' relationships and an independent confirmation of the simulation results. Or, if the permutation aspect of B 's definition makes such an analytical result intractable, it would be interesting to explore the relationship between correlation and variance

in order to understand the driver of the greater difference between models in **Figure 5B** than in **Figure 5A**.

As Reviewer 2 points out (see Comment #31), B is quite closely related to the correlation coefficient. Thus, for clarity, we have refocused this figure (now numbered **Figure 3**) on the relationship between the correlation in global ancestry proportion between mates, $r(x_i, x_j)$ and the variance in global ancestry proportion across individuals, $\sigma_x^2(t)$. This figure demonstrates that relationship between these two variables is not linear, changes over time, and becomes increasingly different between models as $\sigma_x^2(t) \rightarrow 0$.

INTRODUCTION:

12. Please include a brief discussion of within- versus across-group variability, *e.g.*, Lewontin (1972) and related work. Please be mindful of the potential for statements about genetic differences among human populations to be exaggerated or taken out of context.

We appreciate the concern that care must be taken to avoid misstatements (or easily misconstrued statements) about the genetic basis of human variation, particularly with regards to ancestry. We explain the section introducing the social group model (**lines 242-251**): all humans share a common genetic ancestry, the degree of genetic isolation between different groups of humans has been constantly in flux over our evolutionary history, and human “populations” are defined by permeable and socially determined boundaries. These features of human genetic diversity make the definition of “admixture” and “ancestry” fuzzy when it comes to human population genetics.

Empirical analyses of admixture use genetic variation within and between source populations to assign genotyped loci to sources, and thus, infer longer local-ancestry tracts. This makes the accuracy of empirical analyses sensitive to the relative apportionment of genetic variation. In contrast, our simulations directly trace local-ancestry tracts over time from the source populations into the admixed individuals. Indeed, we have not simulated *genotypes* at all – we are completely agnostic to genetic variation within the source populations and the degree of genetic divergence between the source populations. We have added a sentence clarifying that we did not model genotypes in our description of the simulation framework (**line 141**) and raise this point again in the Discussion (**lines 665-670**).

MODELS:

13. The model equations do not match their corresponding plots in **Figure 1**.

- a. The stationary-preference model equation (**line 169**) and the increasing-preference model equation (**line 182**) are both functions of c and x_i , and have an upper bound of 1. The corresponding plots (panels A and B) are a function of α and x_i , and have an upper bound of 2.
- i. Please correct whichever is mistaken or explain if panels A-B are not plotting the equations in **line 169 and 182**, respectively.

We have clarified in the figure caption that **Figure 1** is plotting each model in terms of α , the shared mate-choice bias parameter. This decision is explained in **lines 166-180** and the derivation of α from c and σ^2 are given when each model is introduced.

We have also added a sentence to the **Figure 1** caption explaining that the values of $\psi_{i,j}$ have been scaled for visualization and changed the y-axis of each subplot to read “Scaled $\psi_{i,j}$ ”.

Specifically, we have scaled the values such that $\psi_{i,j} = 1$ means that the probability of individual j being chosen as individual i 's mate is $1/n$, where n is the number of individuals in the population. This is the case under random mating. Thus, $\psi_{i,j} > 1$ means the probability of this mating pair is greater than $1/n$ and $\psi_{i,j} < 1$ means the probability is less than $1/n$.

- ii. Please also clarify if panel B is using the infinite-sites model introduced in **line 184**.

We have removed the statement about the infinite-sites model from the main text because it pertains only to **Figure 1** and not to the increasing-preference model more generally. Thus, we have added a clarification in the figure caption for panel B that we assume $\sigma_x^2(t) = 2^{-t/2}$ following Liang *et al.*

- b. The broad-preference model (**line 197**) has neither α nor c , but instead is parameterized by x_i and “a fixed parameter σ ” which is never defined (**line 196**). The corresponding plot (panel C) again is a function of x_i and α .

- i. Please specify the values of σ used for panel C and correct the figure legend.

As noted above, we have modified the caption of **Figure 1** to state that all models have been re-parameterized in terms of α . We have also added an explanation in **lines 221-228** that the broad-preference model is based on a normal distribution with variance σ^2 , which is a constant (in contrast to the increasing-preference model, which is parameterized in terms of the variance in global ancestry proportion across individuals, $\sigma_x^2(t)$). We also explain the relationship between σ^2 and α on **line 226**.

- c. The social group model does have an α parameter, but the values plotted in panel D do not match the values obtained when you plug $\alpha = 4$ or $\alpha = 10$ into the equation at **line 235**.

- i. Please correct the values in panel D or the equation at **line 235**.

We have modified the y-axis label on each panel of **Figure 1** to say “scaled $\psi_{i,j}$ ” and explained in the figure caption how the scaling was done and why.

- d. Please explain what is meant by “weighted sampling” (**line 258**).

We have changed the text to read: “parent 2 was sampled proportional to mating weight $\psi_{i,j}$ ” (**line 162**).

14. Please number important equations and reference those numbers in addition to the model names.

We have numbered all equations. Those equation numbers are referenced in the caption of **Figure 1**.

15. Please consistently define all model variables when they are introduced. See Major Comments 4 and 5.

Please see our response above. (We have re-numbered the reviewer comments to maximize clarity in referencing them. Major Comments 4 and 5 are now #8 and #9.)

16. It seems that much previous work relied on a framework with two distinct sexes, with one being more selective. Please mention (either here or in the discussion) if you expect your simulation results to depend on this choice.

Prior work that focused on sexual selection (*e.g.*, Lande 1981; Seger 1985) considered sexually dimorphic traits. Global ancestry proportion, on the other hand, does not differ in its phenotypic presentation between males and females and the mate-choice models from the literature that we consider here are not sex-specific.

On a more practical note: SLiM can simulate female mate choice, not but male mate choice. Thus, if we had included two sexes in our framework, each mating event would have been composed by a female sampled uniformly from the population and a male sampled proportional to $\psi_{i,j}$ (as defined by the relevant mate-choice function). Since we are not modeling sex-specific traits, we would not expect any impact on the results, outside of the effect of reducing the effective population size by half.

We have added a discussion in **lines 157-160** of the fact that SLiM only allows for female mate choice and that by modeling hermaphrodites we allow all individuals to exercise mate choice. We also mention asymmetries in mate choice as a future direction for model development (**line 674**): there are, of course, numerous plausible ways that mate choice might interact with sex and global ancestry proportion in a more complex model.

17. Please discuss (either here or in the discussion) how the results are shaped by the genetic diversity within populations and the similarity across source populations.

Genetic diversity within populations and the degree of divergence between source populations would impact our ability to accurately call local ancestry tracts and calculate the global ancestry proportion of individuals. However, because we are using simulated data, we have perfect knowledge of the position and length of local-ancestry tracts, and thus, global ancestry proportion. We add a line to the discussion on ancestry estimation error for empirical analyses (**lines 669-670**).

18. **Lines 136-141**: Please explain how x_i and s_i are derived from the SLiM simulated genome.

- a. Please also clarify if the models assume the presence of only two ancestry groups, with proportion $(x_i, 1 - x_i)$ respectively in individual i .

We have added an explanation of how x_i and s_i are derived from the simulated genomes in **lines 142-147**. We explain that there are two source populations in **line 134**.

19. **Lines 170-178**: Why do you choose a focal individual x_1 and compute ψ_i , rather than computing a more general, pairwise weight $\psi_{i,j}$?

The description of how mating pairs are formed in the original version of the manuscript is most accurate to how it works under the hood in SLiM: for each mating event, an individual is selected as mate 1, then a vector of sampling weights is calculated for all potential mates, and finally, mate 2 is selected proportionally to its weight.

However, we agree with the reviewer that it reads better if we instead phrase it as the probability of mating between individuals i and j is proportional to $\psi_{i,j}$ – and this phrasing does not substantively change the meaning. We have revised **lines 152-165** and **Figure 1** accordingly.

20. **Lines 179-190**: Please define the generation-specific population variance in ancestry, σ_g^x , in terms of both g and x . Is the x here the mean ancestry proportion across the population at generation g ?

We have revised our description of the increasing-preference model in **line 214** to say that “we re-scale c in each generation t using the variance in global ancestry proportion observed in that generation, $\sigma_x^2(t)$ ”. We have also made sure to consistently use the phrase “global ancestry proportion, x ” to remove any ambiguity about what the symbol x refers to.

21. **Line 196**: Please define the “fixed parameter, σ ”. If σ is not related to the variance in ancestry, please use a different symbol for this parameter.

We have improved our description of the broad-preference model (**lines 221-227**) to explain that the parameter σ^2 defines the shape of the normal distribution. See Comment #13(b).

22. **Lines 207-210**: Please clarify whether the categorical trait in these examples is species identity.

We have added the word “identity” to the sentence “Mechanistically, this can be construed as a model of biased mating by species identity or source population” (**line 241**).

23. **Lines 231-235**: As discussed in Major Comment 5, please include the permissible range of α and ψ_i . Please also include the cases when ψ_i is minimized or maximized, and the interpretation of these cases (e.g., “when $\alpha = 0$, ψ_i takes its maximal value of 1, implying that...”).

We have better explained the interpretation of α and its permissible range in **lines 173-177**. We have also clarified that it is the value of $\psi_{i,j}$ relative to other individuals matters for mate choice.

24. **Lines 267-284:** The purpose of this subsection is unclear. How does this section and its α parameter relate to the previous models and their c , σ , and α parameters? Why is the $r_{t=1}$ notation used here but nowhere else, and is this the same correlation discussed elsewhere? If this subsection's purpose is to provide an empirical definition of the mate choice strength parameter used in the earlier models, then this section should be included at the beginning of the models section rather than at the end.

We have moved this subsection (**lines 166-180**) to come before the details of the four models, rather than after. We hope that this will alleviate many points of confusion for the reviewer as addressed in previous comments.

- a. **Line 268:** please include parenthetical references to the mate choice function equation numbers.

We have added a parenthetical reference to the mate choice function equations in **line 167**.

- b. **Lines 275-280:** I appreciate this explanation of the mating bias parameter, α , but it needs to be far earlier in the models section (e.g., **lines 231-235**, or earlier if it applies to the c parameter used in the continuous models as well).

Per the reviewer's suggestion, this subsection now comes before the mate-choice functions in **lines 166-180**.

- c. **Lines 280-284:** This is quite an abrupt introduction to $r_{t=1}$.
 - i. Please ease the reader into this explanation by inserting a preceding sentence that defines and motivates the correlation in mate ancestry.
 - ii. Please provide more intuition for the equation at **line 284**. For example, at the moment, it is unclear where the 0.25 is coming from.
 - iii. Why do you introduce the notation for $r_{t=1}$ here but not use it anywhere else?

We have removed the equation for $r_{t=1}$ because it was distracting from the main point that we wished to get across here: "While our models differ in their dynamics over time, simulations with the same α have the same correlation in global ancestry proportion between mates, $r(x_i, x_j) \in [0,1]$, for the first generation post-admixture under all models" (**lines 177-180**).

RESULTS:

25. **Lines 327 - 368:** Please describe the simulations' initial ancestry proportion distribution.

We have improved our description of the initial ancestry proportion distribution and how it changes over time in **lines 404-409**. We also direct the reviewer to **Figures S2, S3, S4, S5, and S11**.

26. **Figure 3:** Would it be helpful to plot the line $y = x$ on these plots as a guide to the eye? Also, I assume these plots should be symmetric about $y = x$; are asymmetries just due to sampling noise?

We have added a $y = x$ line to this figure (now numbered **Figure 4**). Under the increasing-preference model, the asymmetry about $y = x$ is simply due to sampling noise. There is greater asymmetry under the social group model because mate-choice is not based on global ancestry proportion: due to compounding effects of sampling noise over 20 generations, there are more individuals in social group A (the cluster with lower global ancestry proportion) than in social group B in the simulation shown in panels B and D.

27. **Lines 415-433:** Please define $\bar{\Delta}_x$ and $\bar{\Delta}_{permuted}$. As discussed above, please also consider expanding this section if B is intended to be used as a summary statistic to understand assortative mating in real data.

We have removed the discussion of B and related statistics because the results are so similar to those using the correlation in global ancestry proportion between mates, $r(x_i, x_j)$. We do still reference $\Delta_x = |x_i - x_j|$ and why it is a useful thing to visualize in **lines 440-444** and **Figure S21**.

Reviewer #2:

This paper is a simulation study of various models of ancestry-based mate choice in admixed populations. The models are in two classes: "ancestry similarity" models where the probability of mating depends directly on the similarity of individuals' ancestries, and a "social group" model where individuals are more likely to mate if they are in the same social group (with distinct ancestries initially occupying separate social groups).

The strength of assortative mating is usually measured by the correlation coefficient between mates. The authors find that commonly used ancestry-similarity models of assortative mating cannot maintain a fixed positive ancestry correlation between mates in the generations after admixture. The reason is that, after admixture, recombination rapidly mixes together the two ancestries in the population, reducing variance among individuals in their ancestry fractions. As individuals become more ancestry-similar, they become less discriminatory in mate choice under these standard models. As the authors show, the situation can be rescued by an "increasing-preference" ancestry-similarity model in which the strength of the preference increases as the average ancestry differences among individuals decline---i.e., individuals become more discriminatory in mate choice.

The authors also show that a given ancestry correlation coefficient among mates can be achieved via both ancestry-similarity and social-group models; this finding complicates mechanistic interpretations of such correlation coefficients.

In general, I found the results in the paper interesting, and they certainly add some nuance to how we commonly think of measuring and understanding assortative mating. I have one major comment on the authors' key findings, and a few more minor comments, all detailed below.

MAJOR COMMENT:

28. The authors' primary goal is to aid our understanding of the etiology and consequences of ancestry-based assortative mating in humans. However, the mating structure that they implement using the `mateChoice` callback in their SLiM simulations allows individuals to mate an unlimited number of times within a generation (see **lines 142-147**). In reality, the variance across humans in their reproductive success is more limited.

This difference between the authors' model and the mating structure of human populations could potentially cause large differences in ancestry dynamics under various kinds of assortative mating. The reason is that, with polygamous mating, models of assortative mating can in some cases induce strong sexual selection in favor of one ancestry or the other. In the authors' simulations, neither ancestry can be systematically favored by sexual selection because they start as equal fractions of the population, but the possibility of sexual selection then makes this (unrealistic) assumption of precisely equal starting fractions crucial.

So I think it is important for the authors to show that their results are not quantitatively driven by the potentially large variances in reproductive success among individuals that can arise in their model. The simplest way to do this would be to enforce monogamy in the simulations, with

each individual mating only once, and each mating pair producing two offspring. This scenario is possible, if probably a little tedious, to simulate using the non-Wright-Fisher environment in SLiM (see, *e.g.*, Veller & Coop, 2024). I confess that I am not entirely sure how to port the authors' functional forms for mate choice to this monogamous scenario, in which mates must be picked without replacement rather than with replacement but given that monogamy is a more realistic scenario for many human populations than the authors' scenario of unlimited polygamy, I think it is an important exercise to carry out.

(Alternatively, the authors could (i) directly measure the variance in reproductive success among individuals in their simulations, and check if these are very dissimilar to values measured in humans, and (ii) carry out some simulations with unequal starting fractions of the two ancestries, to see if sexual selection is induced in such cases.)

We thank the reviewer for raising this important and interesting question. While it is possible to simulate non-Wright-Fisher populations in SLiM, it is an entirely different modeling framework, in which we would need to consider the impacts of varying additional parameters, including carrying capacity and birth/death rates. Modeling monogamous mating, for instance, would require us to consider many additional modeling decisions: How much does per-pair reproductive success vary? Is everyone strictly monogamous or should we incorporate some rate of extra-pair paternity? Is one sex choosy and the other passive, or do both partners participate in a mutual mating decision? Does every individual contribute to the next generation?

One particularly thorny consideration when sampling without replacement is the order in which individuals choose mates. We cite Xie *et al.* 2015, who demonstrate through simulation that allowing “high-status” individuals to choose mates first will result in assortative mating along the entire spectrum of status levels, even if all individuals would prefer a high-status mate, because low-status individuals have fewer potential mates left to choose from (**line 74**). This suggests that combining our mate-choice functions with a biased sampling order would accentuate the observed signatures of assortative mating. Indeed, in preliminary analyses performed using a simple python script, we found that allowing individuals with greater source 1 global ancestry proportion to choose mates first produced a higher $r(x_i, x_j)$ than randomizing the order in which individuals choose mates. While it is possible to come up with a plausible justification for sampling either in a random or a non-random sort order, that choice is not within the scope of our model-comparison objective.

As we state at the end of the paper, “more complex models are likely to be most useful when targeted to matching the parameters of a population of interest whose history is well understood and less well suited to a general exploration of parameter space” (**lines 678-680**). We have added a note about the increased complexity of these models to the Discussion (**lines 645-647**) and added monogamous mating as a future direction in **line 675**.

Per the reviewer’s suggested alternative, we have added in two supplemental figures to address the concern that biased non-monogamous mating has led to sexual selection in our simulations. First, **Figure S14** shows that variance in reproductive success is limited in our simulations, even when the initial contributions of the two source populations are unequal: across simulations, we find that individuals in generation $t = 20$ participate on average in one mating event as parent 1 and one mating event as parent 2, and 86% of individuals participate in 3 or few mating events total. Second,

Figure S15 demonstrates that the global ancestry proportion of an individual has limited impact on their reproductive success.

MINOR COMMENTS:

29. **lines 151-153**: "... (Lande 1981; Kirkpatrick 1982; Seger 1985). Generally, under these models one locus controls the male phenotype while a second locus controls the female preference for that phenotype." This is not true for quantitative genetic models of sexual selection such as the cited Lande 1981 paper and the large literature it inspired.

We thank the reviewer for pointing out that we had not stated the distinction we wished to draw in this paragraph precisely enough: we are distinguishing between (1) models that consider male phenotype and female preference as distinct traits with distinct genetic bases, and (2) models in which females prefer males with reference to her own phenotype, linking the genetic basis of male phenotype and female preference. The text now reads (**lines 182-189**):

"Individual-based mate preference models originate in the sexual selection literature, where they were developed for studying the co-evolutionary dynamics of male secondary sex characteristics and female mate choice (Lande 1981; Kirkpatrick 1982; Seger 1985). Generally, under these models, male phenotype and female preference are controlled by distinct genetic loci (or sets of loci). In the speciation literature, an alternative family of models are concerned with assortative mating as a mechanism of sympatric speciation and model mate preference based on phenotypic similarity between mates for some ecologically relevant trait (Dieckmann & Doebelli 1999; Burger & Schneider 2006; Pennings *et al.* 2008; Rettelbach *et al.* 2013)".

30. **line 181**: Why σ for the variance, and not σ^2 as usual?

We agree with this reviewer that it should be σ^2 and have revised the text to consistently refer to this quantity as the "variance in global ancestry proportion across individuals at time t , $\sigma_x^2(t)$ ". This is first defined on **line 215**.

31. **lines 415-433**: To complement the usual metric for the strength of assortative mating, the correlation coefficient among mates, the authors suggest a new statistic which they call B . B is based on the average absolute distance in ancestries among mates, $|x - y|$; specifically, it is $1 -$ [the ratio of the average value of $|x - y|$ under assortative mating to its average value if mate choice were random, as computed by the authors using a bootstrap] (line 430). However, this statistic B is mathematically almost the same thing as the correlation coefficient, as can be seen if we compute the same metric but replace the absolute value ($|x - y|$) with the square $((x - y)^2)$.

Call this analogous metric B' . Under assortative mating:

$$E[(x - y)^2] = Var(x) + Var(y) - 2Cov(x, y) = 2[Var(x) - Cov(x, y)]$$

since x and y are identically distributed.

Under random mating, $Cov(x, y) = 0$, so

$$E[(x - y)^2] = Var(x) + Var(y) = 2Var(x)$$

So,

$$B' = 1 - \frac{[Var(x) - Cov(x, y)]}{Var(x)} = \frac{Cov(x, y)}{Var(x)} = Cov(x, y) / \sqrt{Var(x)Var(y)}, \text{ the correlation coefficient } \rho(x, y).$$

This close mathematical correspondence between B and the correlation coefficient presumably explains why they behave almost identically in the authors' simulations (e.g., **Figure 5A**).

This point is well taken. In response, we have removed the discussion of B from the manuscript altogether. In the referenced figure (now numbered **Figure 3**), we instead examine the relationship between correlation in global ancestry proportion between mates, $r(x_i, x_j)$, and variance in global ancestry proportion across individuals, $\sigma_x^2(t)$, directly. We demonstrate that this relationship is not linear, changes over time, and becomes increasingly dissimilar between models as $\sigma_x^2(t)$ approaches zero. We have replaced our discussion of B with text pointing out that visualization of the distribution of $\Delta_x = |x_i - x_j|$ can provide more information than summarizing these data in the correlation coefficient (**lines 440-444; Figure S21**).

32. **Figure 7**: I found this figure unnecessarily difficult to understand. There are no x-axis labels in panels A and B, and I can't actually tell what these should be. There are two sets of dashed lines in each panel A and B, but only one is referred to in the caption (in general, these panels would benefit from direct labelling rather than just descriptions in the caption). Also, could A and B not be in a single panel with the same axes? Panels D and E should are not described as such in the caption; instead, their description is contained in that of panel C.

We apologize for the inadvertent inclusion of an incomplete version of **Figure 7**. We have improved the figure (now numbered **Figure 6**) and its caption per the reviewer's comments. This includes properly labeling the x-axis for panels A and B and revising the caption to explain what each line represents. We have also changed the label in the caption to refer to panels C-E.

33. **lines 571-573**: "global ancestry proportion ... is determined by every locus in the genome, meaning that the entire null distribution of any parameter of interest is likely to be affected [by assortative mating]". I didn't understand this.

This sentence (**lines 575-578**) has been revised to read: "Furthermore, global ancestry proportion is an unusual quantitative phenotype, in that its trait value is integrated across every locus in the genome. How this genome-wide involvement might impact analyses, such as selection scans and association studies, that attempt to distinguish implicated loci from neutral loci remains unclear".

34. **lines 595-597**: This seems unfair to the cited papers (one of which is co-authored by the senior author of the present manuscript!). Those papers are, in fact, correct to "assume that [biased-mate choice] models are sufficient to generate a correlation in ancestry", and, though they

would have been incorrect to assume that the models are sufficient to generate a constant or persistent correlation in ancestry, they did not, as far as I can tell, make such an assumption.

We have revised this statement to focus on the relationship between the correlation in global ancestry proportion between mates and the variance in global ancestry proportion across individuals, and how that relationship changes over time. It now reads (**lines 599-605**):

“Prior studies that have modeled biased mate choice to develop theory about assortative mating have either not considered how the decrease in variance in global ancestry proportion across individuals over time impacts the correlation in global ancestry proportion between mates (Kim *et al.* 2021; Muralidhar *et al.* 2022) or have explicitly modeled a constant correlation coefficient in the face of decreasing variance (Huang *et al.* 2024). Our work highlights that variance in global ancestry proportion across individuals plays an essential role in determining whether a positive correlation can be observed and, furthermore, whether mating is effectively random”.

TYPOS/WORDING:

35. **line 63**: “have suggested temporal structure” → “have suggested that temporal structure”

This typo on **line 74** has been corrected.

36. **line 435**: “that correlation coefficient” → “that the correlation coefficient”

This sentence has been removed.

37. **line 455**: “produced in an excess” → “produces an excess”

This typo on **line 456** has been corrected.

38. **line 630**: “patterns ... is” → “patterns ... are”

This typo on **line 636** has been corrected.

January 27, 2025

RE: GENETICS-2024-307741

Dr. Amy Goldberg
Duke University
Evolutionary Anthropology
Durham, NC

Dear Dr. Goldberg:

Congratulations! We are delighted to inform you that your manuscript entitled "Differentiating mechanism from outcome for ancestry-assortative mating in admixed human populations" is acceptable for publication in GENETICS. Many thanks for submitting your research to the journal.

The reviewers had a few suggestions for improving the manuscript that you may want to consider. You can view their comments at the bottom of this email.

To Proceed to Production:

1. Format your article according to GENETICS style, as discussed at <https://academic.oup.com/genetics/pages/general-instructions>, and upload your final files at <https://genetics.msubmit.net>.
2. Your manuscript will be published as-is (unedited-as submitted, reviewed, and accepted) at the GENETICS website as an Advanced Access article and deposited into PubMed shortly after receipt of source files and the completed license to publish. Please notify sourcefiles@thegsajournals.org if you do not wish to publish your article via Advanced Access.
3. We invite you to submit an original color figure related to your paper for consideration as cover art. Please email your submission to the editorial office or upload it with your final files. You can submit a small-sized image for evaluation, and if selected, the final image must be a TIFF file 2513px wide by 3263px high (8.375 by 10.875 inches; resolution of 600ppi). Please avoid graphs and small type.

If you have any questions or encounter any problems while uploading your accepted manuscript files, please email the editorial office at sourcefiles@thegsajournals.org.

Sincerely,

Simon Gravel
Associate Editor
GENETICS

Approved by:
Nicholas Barton
Senior Editor
GENETICS

note: Please add jnls.author.support@oup.com and genetics.oup@kwglobal.com (or the domains @oup.com and @kwglobal.com) to your email program's "safe senders" list. You will be contacted by both at various points during the production process.

Review comments (if applicable):

Reviewer #1 :

Thank you to the authors for their thorough response to my comments. I appreciate the refined focus of the results section and the improved clarity of the manuscript, figures, and equations. In particular, I think the goal of the manuscript is now more clear,

and it is also more apparent that the manuscript achieves that goal. I think this paper will serve as a nice reference for readers seeking to better understand existing models of ancestry-assortative mating.

Reviewer #2 :

The authors have revised their manuscript in a way that has substantially improved its clarity, and have addressed all of my previous comments very thoughtfully in the process.

Some minor bookkeeping suggestions below:

- Eq. 1 and lines 170-176: The interpretation of these values of alpha seem to require that the populations admix in equal proportions, which, if so, should be stated.
- line 380: space after "higher"
- line 508: "model"  "models"